# Attraction to similar options: The Gestalt law of proximity is related to the attraction effect

Liz Izakson [1,2]*, Yoav Zeevi[1,3], Dino J. Levy[1,2]

**1** Sagol School of Neuroscience, Tel Aviv University, Tel Aviv, Israel, **2** Coller School of Management, Tel Aviv University, Tel Aviv, Israel, **3** Department of Statistics and Operations Research, Tel Aviv University, Tel Aviv, Israel

* lizizakson@mail.tau.ac.il

## Abstract

Previous studies have suggested that there are common mechanisms between perceptual and value-based processes. For instance, both perceptual and value-based choices are highly influenced by the context in which the choices are made. However, the mechanisms which allow context to influence our choice process as well as the extent of the similarity between the perceptual and preferential processes are still unclear. In this study, we examine a within-subject relation between the *attraction effect*, which is a well-known effect of context on preferential choice, and the Gestalt law of proximity. Then, we aim to use this link to better understand the mechanisms underlying the *attraction effect*. We conducted one study followed by an additional pre-registered replication study, where subjects performed a Gestalt-psychophysical task and a decoy task. Comparing the behavioral sensitivity of each subject in both tasks, we found that the more susceptible a subject is to the proximity law, the more she displayed the *attraction effect*. These results demonstrate a within-subject relation between a perceptual phenomenon (proximity law) and a value-based bias (*attraction effec*t) which further strengthens the notion of common rules between perceptual and value-based processing. Moreover, this suggests that the mechanism underlying the *attraction effect* is related to grouping by proximity with attention as a mediator.

## Introduction

All of our decisions, from simple ones like the size of the popcorn we choose to buy in the cinema to more complicated ones like choosing our life partner, are influenced by other available alternatives (as well as unavailable ones) in the environment. Other available or unavailable alternatives in the current environment of the choice set are considered spatial context. A well-known example of the effect of spatial context is the *attraction effect* [1,2]. Suppose you are choosing between a small-sized popcorn that is relatively cheap and costs only $3 (*competitor*) and a large-sized one which costs $6.5 (*target*). In this scenario, no option has a clear advantage over the other. The small-size option is better in one attribute (price), while the large-size option is better on the other attribute (size). Now imagine a third option of popcorn that is medium-sized and costs $7 (*decoy*). Under these circumstances, the *decoy* is *asymmetrically dominated*, since it is inferior to the *target* option in both attributes (size and price), but

**Data Availability Statement:** All data is available from https://osf.io/jzk6y/.

**Funding:** D.J.L. received funding from the United States-Israeli Bi-national Science Foundation (CNCRS = 2014612). https://www.bsf.org.il/. The

funders had no role in study design, data collection and analysis, decision to publish, or preparation of the manuscript.

**Competing interests:** The authors have declared that no competing interests exist.

inferior to the competitor in only one attribute (price). Numerous experiments have shown that the presence of such a *decoy* in the choice set shifts preferences toward the *target* option [1,3–5].

The *attraction effect*, as well as other *decoy* effects, such as the similarity and compromise effects [2,6], violate integral axioms of normative theories of choice. These include the independence of irrelevant alternatives (IIA) axiom (the introduction of an irrelevant option to a choice set should not change the preference between existing options [7] and the regularity principle [6].

These various *decoy* effects and other related context-dependent phenomena, such as framing effects (e.g. asymmetries in valuation of gains and losses) [8,9] demonstrate the importance of the specific context that is experienced by the decision maker during choice to the valuation process. Although context is an integral part of our decision-making process, the mechanisms underlying these phenomena are still unclear. Understanding the mechanisms which allow context to influence our valuation and choice processes can shed light on human choice mechanisms in general, and the generation of values in a complex environment in particular.

Several explanations have been proposed to account for the change in preference induced by different context effects. Specifically, many suggested computational models use the notion that people accumulate evidence for alternatives over time, and make a choice when the evidence reaches a decision criterion [10–14]. According to one of these sequential sampling models, the multi-alternative decision by sampling (MDbS) model, the accumulation of the evidence is made by pairwise comparisons on a single attribute [14]. Importantly, the more similar the attributes of different options are, the more time the observer would spend comparing between them. In our example, the *decoy* (a medium-sized popcorn which costs $7) and the *target* (a large-sized popcorn which costs $6.5) options are the most similar pair on both attributes (size and cost) between the three available pairs. Therefore, according to the MDbS model, the duration of comparison between them would be longer. This prediction has been supported empirically using eye movements [15]. Moreover, the higher the probability to compare between a specific pair of options, the higher the probability to choose the better option between this pair [14]. In our example, a comparison between the *decoy* and the *target* would yield that the *target* is the better option in both attributes (size and cost), thus the probability to choose it would be higher. Therefore, an important part of the mechanism underlying the *attraction effect*, according to the MDbS model, is the perception of the differences and similarities between two options in the attribute space. Other sequential sampling models also ascribe a crucial role to the distance between options in the mechanism which leads to the *attraction effect*. For example, according to the Multi-alternative Decision Field Theory (MDFT), lateral inhibition is increased when the options are closer to each other in the attribute space [16] and according to the Multi-Attribute Linear Ballistic Accumulator (MLBA) model, options that are more difficult to discriminate (i.e., more similar options) receive more attention, thus increasing the probability of the more dominant option to be chosen, which leads to the *attraction effect* [17]. Note that the explanation for why people drive more attention to similar options is still unclear. In the current study, as our attributes are monetary amount and winning probability, we refer to the distance between options in the attribute space as *value distance* (VD).

This raises an intriguing possibility. Is the perception of *value distance* similar or analogous to the way we perceive and are affected by actual *physical distance*? There are many similarities and analogies between sensory and value processing. First, computational models, in both sensory perception and value processes, use transformation of information from objective magnitudes to a subjective scale in order to explain subjects' performance. In perception this is known as the Weber-Fechner function [18], which states that the increase in the perceived

magnitude of a certain stimulus declines as the stimulus intensity increases, and in value-based choice it is known as the Bernoulli function, or the utility function [19], which describes the notion that the marginal utility of a certain object declines as the total amount of this object increases. Second, value modulation was observed in both visual cortex [20] and auditory cortex [21] in a modality-specific way [22]. Moreover, recent studies have shown that visual selective attention is also driven by the learned value of a stimulus [23,24]. Third, in recent decision models, the mechanisms for why people demonstrate violations of normative axioms such as the IIA is explained using the analogy of perceptual biases [25–28]. According to this view, similar to perceptual illusions, both spatial and temporal, choice biases (or cognitive illusions) are the result of inherent cognitive and neurobiological limitations in both the capacity and the efficiency of information processing. For example, a recent study by Khaw and colleagues [29] demonstrated that value is influenced by adaptation in a similar way as perception: subjects' valuations were lower after high-value adaptations and higher after low-value adaptations demonstrating a repulsive effect of recent values. By the same token, several studies have shown that the *attraction effect* as well as other *decoy* effects (e.g., similarity, compromise) emerge in simple perceptual decision-making tasks [30,31]. Therefore, we propose that we can use the vast knowledge that we gained regarding sensory processing, in order to better understand the mechanisms underlying choice biases such as the *attraction effect*.

In the current study, we examined a potential link between the *attraction effect* and the Gestalt law of proximity and then used this link in order to better understand the mechanisms underlying the *attraction effect*. The proximity law suggests that we tend to combine elements that are close to each other and treat them as one group [32]. We chose specifically this law because it refers to the *physical distance* between objects, and as was mentioned above, an integral feature of the *attraction effect* is the position of the *decoy* relative to the *target* which defines the distance between them [1]–what we refer to as the *value distance*.

Because there are many similarities and analogies between sensory processing and value processing, we hypothesized that the *value distance* between a *decoy* and a *target* option would be conceptually analogous to the *physical distance* between objects as formulated by the Gestalt law of proximity. According to Koffka [1], the bigger the physical distance between objects, the less chance there is to perceive these objects as grouped by proximity. Thus, we hypothesized that the bigger the *value distance* between the *target* and the *decoy*, the less the subject would be affected by the *attraction effect* [33].

Furthermore, we hypothesized that the larger the sensitivity to grouping by proximity a subject will have, the larger will be her susceptibility to the *attraction effect* (the subject's probability to choose the *target* would be higher), because it would be easier for the subject to perceive the similarity between the *decoy* and the *target*, since she would more readily group them together.

To address these questions, we performed two independent experiments, where we pre-registered the replication experiment according to the results of the first experiment in Open Science Framework (https://osf.io/jzk6y/). In both experiments, all subjects performed a Gestalt-psychophysical task and a decoy task. Thereafter, we compared, for each subject, their behavioral sensitivity in both tasks. As we will present below, we found that the more sensitive a given subject is to the proximity law, the more she displayed the *attraction effect*. Therefore, we will illustrate how the proximity law might account for the *attraction effect* with attention as a mediator.

## General methods

### Data sharing

Our pre-registration forms as well as all of our data and codes are shared on Open Science Framework (https://osf.io/jzk6y/).

**Table 1. Demographic information.**

| | Sample size (excluded) | Females (percent) | Age M (SD) |
|---|---|---|---|
| Experiment 1 | 52 (14) | 29 (56%) | 26.53 (4.12) |
| Replication | 102 (21) | 58 (57%) | 25.63 (5.39) |

## Subjects

A total of 119 healthy subjects took part in this study; each participated in one of two identical experiments. Experiment 1, *n* = 52; Replication, *n* = 102; see Table 1 for a demographic description of each experimental sample. We pre-registered the replication experiment according to the results of Experiment 1 (https://osf.io/jzk6y/). The replication experiment was identical to Experiment 1, and aimed to validate its results. We performed a power analysis with the data obtained from Experiment 1, which yielded a minimal *n* = 74 for detecting the Gestalt threshold effect on choice with 80% power and alpha = .05. (Power was calculated using a Wald test, examining the significance of the Gestalt threshold variable in a mixed-effect logistic regression). However, based on Experiment 1, we assumed that some of the subjects (~20%) will be excluded from the study based on our strict exclusion criteria (available online at https://osf.io/jzk6y/). Therefore, we chose to pre-register a larger sample size of *n* = 100 for the replication experiment.

In both experiments, all subjects performed three tasks: first, a calibration task for the forthcoming decoy task, second, a Gestalt-psychophysical task and third, a Decoy task. The experiment was conducted in the laboratory on monitors with screen resolutions of 1,920 × 1,080 pixels. All subjects received a participation fee and were also paid according to their winnings in the experiment. They all signed a written informed consent that was approved by the ethics committee at Tel Aviv University (ethics approval number 0000080–1).

## Stimuli & procedure

**Calibration task.** In the calibration task, for each subject we estimated two indifference points that served as the basis for generating the stimuli used in the Decoy task. Subjects made repeated choices between a choice option with 61% chance to win 22 NIS (and a 39% chance to win zero)–option A, and an option with *p* chance to win 42 NIS (and 1-p to win zero)–option B. The numbers were randomly jittered across trials by ±1 or ±2 to prevent subjects from memorizing their choices. We systematically varied the value of *p* from 19% to 52%, resulting in 11 different unique trials. Each trial was repeated 6 times. We calculated the indifference point (or point-of-subjective-equivalence) for each subject based on a logistic regression model. That is, for each subject, we identified the expected-value difference between the two options that corresponds to the 50% point on the y-axis (-constant/slope) according to the logistic model. We repeated this procedure using different amounts and probabilities in order to estimate a second indifference point for each subject. Subjects made repeated choices between an option which offered a 45% chance to win 27 NIS–option A, and another option with a varying chance of *q* to win 59 NIS–option B. As in the previous set of amounts and probabilities, the numbers were randomly jittered across trials by ±1 or ±2 to prevent subjects from memorizing their choices. We systematically varied the value of *q* from 12% to 34%, resulting in 11 different unique trials that were repeated 6 times (The full list of trials for both sets of options A and B is available in S1 Appendix).

In all trials of the calibration task, each trial started with the presentation of a fixation cross at the center of the screen for 500 ms. Thereafter, a table of 2 lotteries was presented at the center of the screen. Subjects were requested to choose their preferred lottery by clicking the

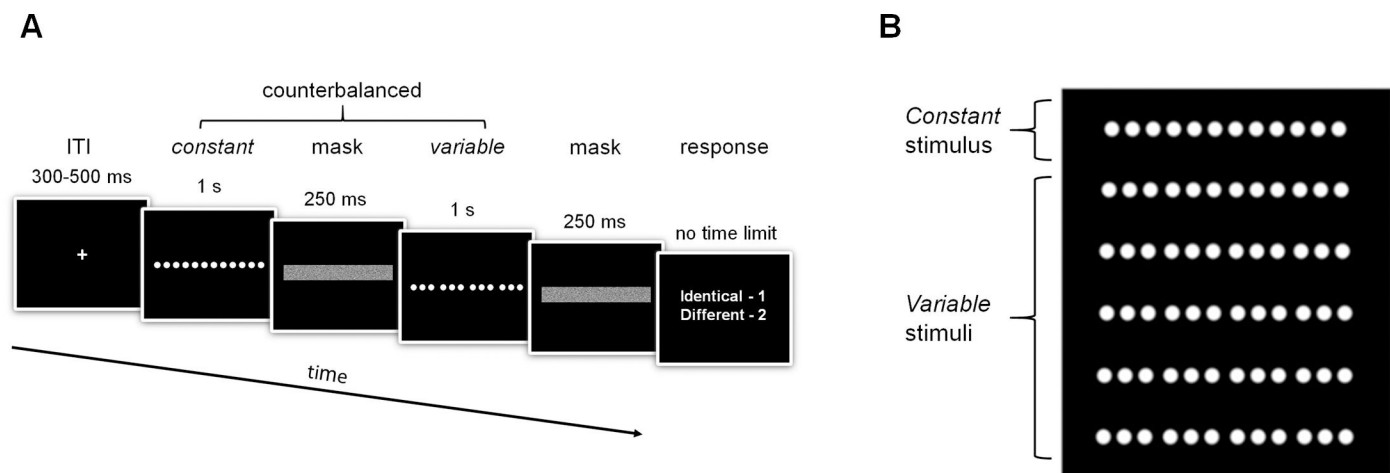

**Fig 1. Gestalt task.** (A) Single trial timeline. (B) Examples of stimulus patterns selected from the set of 29 variable stimuli used in the experiment and a constant stimulus in the Gestalt task. Each individual row represents a different stimulus that was presented separately on the screen in each trial.

number of this lottery on the keyboard. There was no time limit to make a choice. After subjects made their choice, the trial ended with a fixation cross for 500 ms. The two different sets of options A and B were presented in a random order in white text on a black background. The total amount of trials in the calibration task was 132.

The task was incentive compatible. At the end of the task, one of the 132 choices was randomly chosen. The option that was chosen on the randomly selected trial was played out according to the amount and probability of that lottery. The subjects were informed about the results of this lottery only at the end of the experiment (after the end of task 3 –Decoy task). If they won the allotted amount of money of the lottery, it was added to their show up fee.

Based on the two subject-specific indifference points, we generated the choice options for the main Decoy task. This step is important because *decoy* effects are most strongly demonstrated when the *target* and the *competitor* are equally valuable [34].

**Gestalt task.** The aim of the task was to measure, for each subject, the threshold for differentiating between two stimuli of 12 dots arranged in a row. The task is based on the task described in [35]. Fig 1A illustrates an example trial. The subject was presented with a fixation displayed in the center of the screen for 300–500 ms (randomly jittered). Thereafter, a *Constant* stimulus of 12 white dots arranged in a row on a black background was displayed in the center of the screen for 1 second, followed by a Mask (a scrambled picture of the *Constant* stimulus) for 250 ms. The distance between the dots was always constant and was equal to 20 pixels. Then, a *Variable* stimulus of 12 dots arranged in a row was displayed for 1 second in which the distance after the 3rd, 6th, and 9th dot was equally varied across trials (Fig 1B). The second stimulus was also followed by a Mask for 250 ms. The order in which the *Constant* and *Variable* stimuli appeared was randomized across trials. Afterwards, the subject was asked if the two stimuli that were presented (*Constant* vs. *Variable*) were identical or different by clicking '1' or '2' on the keyboard, respectively. There was no time limit for the response phase. In total, subjects observed 29 different *Variable* stimuli in which the spacing distance between each three dots was varied from 20.5 pixels to 34.5 pixels in increments of a half pixel. Each *Variable* stimulus was repeated 6 times (except the trial in which the *Constant* and *Variable* stimuli were identical which was repeated 18 times) for a total of 192 trials.

For each subject, we measured the probability to respond "different" as a function of the increase in physical distance between the dots in order to calculate the threshold to

differentiate between the two stimuli. As was stated above, this task was based on a psycho-physical task presented in [35] where a higher tendency to detect differences in physical distance as well as the tendency to group by proximity is translated to a lower threshold. A subject who is more susceptible to grouping by proximity will detect the differences between the *Constant* stimulus and the *Variable* stimulus at a much lower distance between the triplets of dots, since it would be easier for her to group the row of 12 dots into 4 groups of triplets.

**Decoy task.** The aim of the task was to examine, for each subject, the existence of the *attraction effect*, as well as its strength. We also measured for each subject the influence of the decoy's location (in *value distance*) on the strength of the *attraction effect*. The task is based on the task described in [36]. Fig 2A illustrates an example trial. Subjects performed a series of choices between gambles in three different conditions: (I) *Basic* condition, (II) *Decoy* condition, and (III) *Filler* conditions (Fig 2B).

In the *Basic* condition, subjects made choices between two lotteries (option A and option B). Each lottery had some amount of money to win associated with a winning probability (e.g. 42% to win 30 NIS (and 58% not to win) vs. 30% to win 38 NIS (and 70% not to win)). There were two different sets of options for the *Basic* condition (both of them shown in S2 Appendix). The specific numbers of the amounts and probabilities were randomly jittered across trials by ±1 or ±2 to prevent subjects from memorizing their choices. Importantly, based on the calibration task, the lotteries were individually tailored, such that the subject was close to being indifferent between them. Each *Basic* trial was repeated 8 times for a total of 16 trials.

In the *Decoy* condition, we added a third gamble to the choice set. The additional option, the *decoy*, was either similar to the probability or to the amount of one of the gambles appearing in the *Basic* condition (the *target*). The remaining dimension of the *decoy* (either probability or amount) was parametrically varied in 4 steps, from being 15%, 30%, 45%, and 60% smaller than that dimension in the *target* option resulting in a ranked-order parameter with 4 levels. For instance, if the *target* was a lottery of 21% to win 59 NIS, A *decoy* on the amount attribute which is 15% smaller would be 21% to win 50 NIS (a detailed calculation is provided in S2 Appendix). Additionally, we used two different *decoy* types: range (which gives the target an advantage in its weaker attribute) and frequency (which gives the target an advantage in its stronger attribute). This terminology was introduced in the original article on the *attraction effect* [1].

Therefore, we had 4 different types of trials in the *Decoy* condition (range-probability, range-amount, frequency-amount, and frequency-probability) and 4 different *value distance* (VD) steps, resulting in 16 different *decoy* options for each set of the Basic condition. Each of the *Decoy* trials was repeated 8 times, resulting in 16*2*8 = 256 Decoy trials.

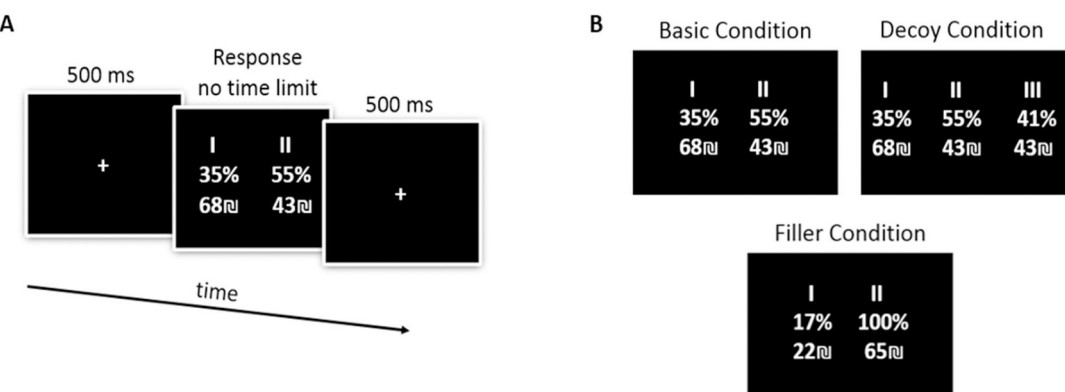

**Fig 2. Decoy task. (A)** Single trial timeline. **(B)** The three task conditions (*Basic*, *Decoy*, and *Filler*).

In the *Filler* condition, there were two types of trials: 1) Binary filler (first order stochastically dominated trials): subjects chose between two options, where one was a non-degenerate lottery (like in the previous conditions), while the other always had a bigger monetary amount and 100% winning probability. We added this condition to validate the continued engagement of the subject with the task throughout the experiment. 2) Trinary filler: subjects chose between three randomly-generated gambling choices not related to the decoy trials. We added this condition to disguise the aim of the experiment from the subjects. There were 8 different *Filler* trials that were repeated 4 times, resulting in 32 *Filler* trials.

In all trials of the Decoy task, each trial started with the presentation of a fixation cross at the center of the screen for 500 ms. Thereafter, a table of either 2 or 3 lotteries (depending on the condition) was presented at the center of the screen. Subjects were requested to choose their preferred lottery by clicking the number of that lottery on the keyboard. There was no time limit to make a choice. After the subjects made their choice, the trial ended with a fixation cross for 500 ms. The different types of trials were presented in a random order in white text on a black background. The total amount of trials in the Decoy task was 16 *Basic* trials + 256 Decoy trials + 32 *Filler* trials = 304 trials.

The experiment was incentive compatible. At the end of the Decoy task, one of the 304 choices was randomly chosen. The option that was chosen on the randomly selected trial was played out according to the amount and probability of that lottery. If subjects won the allotted amount of money of the lottery, it was added to their show up fee.

In order to examine the existence and strength of the *attraction effect* across subjects and steps of VD, we first examined if the probability to choose the *target* was significantly higher than chance level (50%). Then, we used the choice in every trial (*target* or *competitor*) as our dependent variable in order to examine the effect of different predictors (e.g., VD, Gestalt threshold) on the *attraction effect* using mixed-effect logistic regression models. We are aware that there are other measurements for the *attraction effect*, however we chose specifically this one since our main analyses in which we used mixed-effect logistic regressions required a dependent variable per trial (we used the choice in each trial: *target*/*competitor* for each subject). We included analyses of the *attraction effect* using three other measurements that are used in the literature: violation of WASRP (the Weak Axiom of Stochastic Revealed Preference) [39,40], violation of regularity [6,37] and *relative choice share of the target* [38]. We concluded that most of the measurements are very similar to the one we used, and thus, yield similar results (see S1 Text; Robustness analyses for more details).

**Exclusion criteria.** In addition to the reported 119 participants, across the two experiments, 35 additional participants were excluded from final analyses (Experiment 1: 14 subjects were excluded; Replication: 21 subjects were excluded). The exclusion criteria were decided based on Experiment 1, as we preregistered the replication experiment, and then implemented as well in the replication experiment.

Subjects were disqualified due to two exclusion criteria: 1) lack of engagement in the Gestalt task (their slope of the fitted logistic regression was not significantly larger than zero ($p < 0.05$) meaning that they were not sensitive at all to the interval increase between the dots, and 2) they chose "different" more than 50% of the trials that were actually identical meaning that they were biased to answer "different". Subjects were also excluded due to lack of engagement in the Decoy task. That is, if they chose the same option more than 96% of the trials at least in two out of the four blocks of the task, which indicates that they showed no variation in their choices across the different trial types (which is analogous to a low slope in the Gestalt task). We chose the 96% threshold based on a thorough exploration of our data from Experiment 1, and based **all of our** exclusion criteria according to it. These exclusion criteria were listed in the pre-registration of the replication experiment.

**Table 2. Influence of the interval increase between the dots on the propensity to respond "different".**

| Fixed-effects Parameters | Experiment 1 (n = 38) | | | | Replication (n = 81) | | | |
|---|---|---|---|---|---|---|---|---|
| | *B* | *SE*# | *Z* | p-val | *B* | *SE*# | *Z* | p-val |
| Constant | -2.38 | 0.18 | -12.91 | < .001*** | -2.47 | 0.12 | -20.34 | < .001*** |
| Interval increase | 0.34 | 0.03 | 12.81 | < .001*** | 0.38 | 0.02 | 18.69 | < .001*** |
| **Random-effects Parameters** | var | | | | var | | | |
| Constant | 1.12 | | | | 1.01 | | | |
| Interval increase | 0.02 | | | | 0.03 | | | |

We used mixed effect logistic regression with random intercept. The model included a random intercept and slope components, allowing them to interact (using an unstructured covariance matrix specification).

# Robust Std. Err. (Errors clustered by Subject); * p < .05 **p < .01

*** p < .001.

## Experiment 1

### Subjects

38 valid participants completed the three tasks presented in the general methods section (mean age = 26.53, *SD* = 4.12, 29 females; demographic statistics are reported in Table 1). 14 additional subjects were excluded: four performed poorly in the Gestalt task (their slope of the fitted logistic regression was not significantly larger than zero ($p < 0.05$) or they chose "different" more than 50% of the trials that were actually identical). Ten performed poorly in the Decoy task (chose the same option more than 96% of the trials at least in two of the four blocks in the task).

### Gestalt results

In order to examine the influence of the increase in physical distance between the triplets of dots (interval increase) on the propensity of the subject to differentiate between the stimuli, we fitted a mixed effect logistic regression with interval increase as the independent variable, and subject's choices (identical/different) as the dependent variable.

We found that, on average, the propensity to discriminate between the two stimuli increased as a function of the interval increase between the dots demonstrating that subjects were sensitive to the spaces between the triplets of dots ($\beta = 0.34$, p<0.001; left side of Table 2).

Next, we fitted for each subject separately, her behavioral data to a logistic regression with interval increase as the independent variable, and subject's choices (identical/different) as the dependent variable. We then estimated the physical distance between the triplets of dots in which the subject was at chance level (the x value at y = 0.5) based on the best fit logistic function per subject. That is, we estimated the interval increase in which the subject could not tell the difference between the *Constant* and *Variable* stimuli, i.e. the sensitivity threshold. Fig 3A describes the data and the logistic fit of two representative subjects. The subject which is represented by the gray dots has a lower sensitivity threshold in comparison to the subject which is represented by the blue dots. That is, the "gray" subject starts to differentiate between the two stimuli when the physical distance between the triplets is smaller (3.6 pixels) in comparison to the "blue" subject who needs a larger physical distance (9.8 pixels) in order to differentiate between the two stimuli. The top histogram in Fig 3B describes the distribution of sensitivity thresholds across subjects for Experiment 1. The average sensitivity threshold was 7.19 (±0.28) pixels (ranging from 3 to 12 pixels). We then used this variation in sensitivity (termed 'Gestalt threshold') across subjects in order to find a link between the Gestalt thresholds and the tendencies to show an *attraction effect* in the Decoy task.

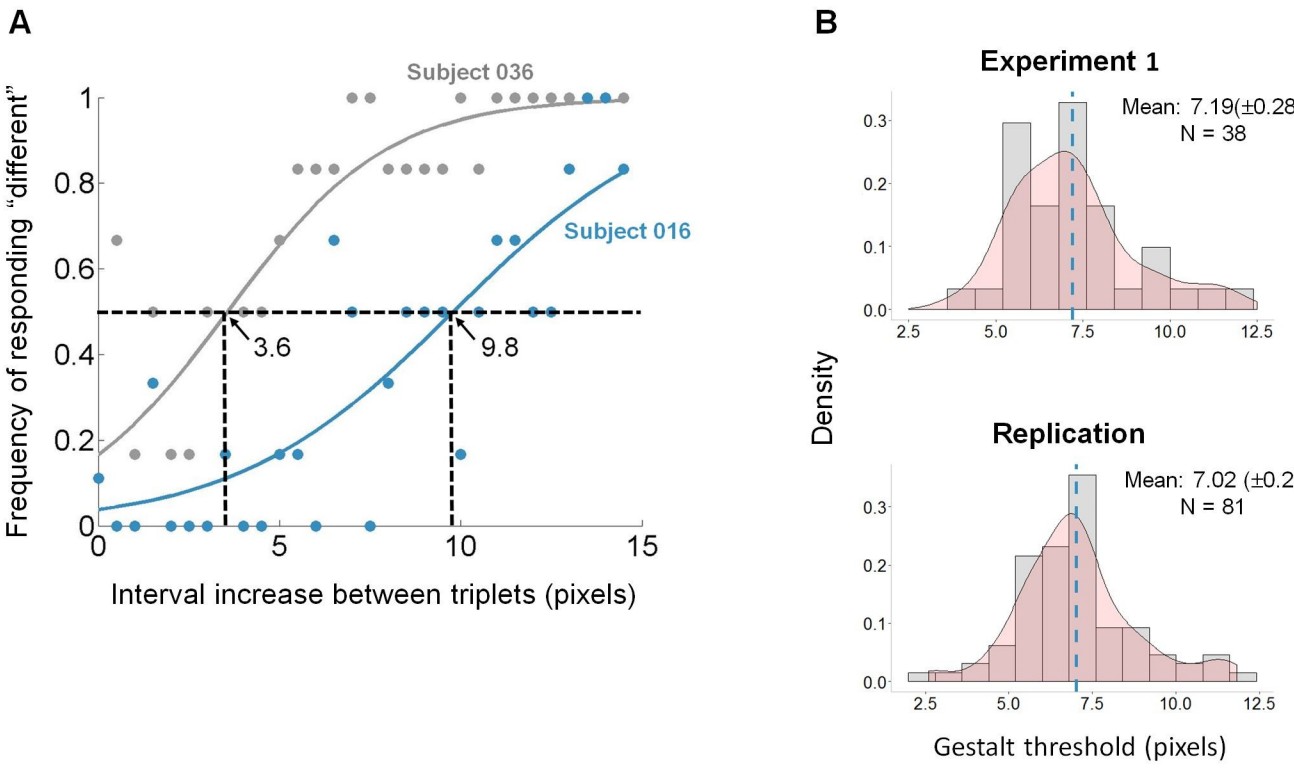

**Fig 3. Gestalt results. (A)** Two representative subjects: the one colored in gray has a lower threshold to differentiate between the two stimuli than the one colored in blue. **(B)** Histograms of the Gestalt sensitivity thresholds calculated for all subjects in Experiment 1 (top) and Replication (bottom). The dashed blue line represents the mean (**Experiment 1:** mean = 7.19(±0.28) pixels; n = 38; **Replication:** mean = 7.02(±0.2) pixels, n = 81).

### Decoy results

**Significant *attraction effect* across subjects, however large heterogeneity between subjects.** In order to examine the occurrence of the *attraction effect* across subjects including all *decoy* locations, we measured the probability to choose the *target* when the *decoy* was asymmetrically dominated by it and compared it to chance level (50%). We found that on average, subjects chose the *target* significantly higher than chance level (one sample t-test, mean = 0.52, CI = [0.51, 0.54], t(37) = 2.76, p<0.01). Although the average effect across subjects is significant, it is a rather small effect. This is probably because there is a considerable heterogeneity across subjects in their probability to choose the *target* (the range of probabilities spreads between 0.38 and 0.68 (Fig 4A)). Therefore, additionally, we examined separately for each subject, the effect of adding a decoy on their probability to choose the *target* option using a binomial test. We found that only ~20% of subjects chose the *target* significantly different than 50% (Experiment 1: 7 out of 38 subjects (18%) chose the target significantly different than 50% (p<0.05); detailed individual results are available in S3 Appendix). While most of the subjects who showed a significant *decoy effect* displayed an *attraction effect*, 29% of them displayed the opposite effect (a *repulsion effect*–higher probability to choose the competitor when the *decoy* was asymmetrically dominated by the *target* [37,38]). These results are in line with previous studies which posited that *decoy effects* are usually weak effects [40,54] and that there are considerable differences between subjects [54].

Moreover, we examined the robustness of our measurement for the *attraction effect* size (choice proportion of *target*) by comparing it with three other measurements that are used in

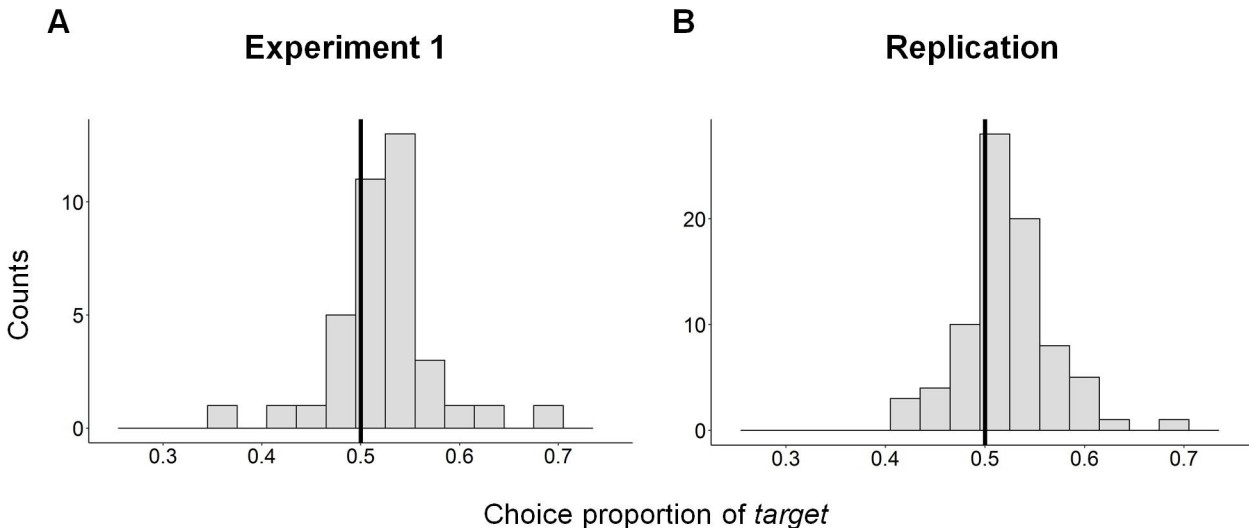

**Fig 4. Histogram of the probability to choose the _target_.** The black line represents choice probability of chance level (0.5). Choice probability of less than chance level (on the left side of the zero line) represent subjects who displayed a general _repulsion effect_, while the choice probability of more than chance level (on the right side of the zero line) represent subjects who displayed a general _attraction effect_. **(A) Experiment 1:** mean = 0.52, n = 38. **(B) Replication:** mean = 0.53, n = 81.

the literature: violation of WASRP (the Weak Axiom of Stochastic Revealed Preference) [39,40], violation of regularity [6] and _relative choice share of the target_ [38]. We concluded that most of the measurements are very similar to the one we used, and thus, yield similar results [detailed information and analyses are provided in S1 Text].

**The influence of the _value distance_ on the probability to choose the _target_.** In order to examine the influence of the _value distance_ on the choice proportion of the _target_, we used the VD, in each trial of the _Decoy_ condition of the Decoy task, as our predictor and subjects' choices (_target_ or _competitor_) as the dependent variable. We first fitted a random-intercept logistic regression model and clustered the errors per subject.

The VD had a significant negative effect on the choice proportion of the _target_ (β = -0.25, p<0.05; left side of Table 3). That is, the further away the _decoy_ was from the _target_ (regardless of the specific attribute (probability/amount) which differentiated between them), the less the subject chose the _target_, and hence, the lower was the _attraction effect_.

However, when we examined the slope coefficients of each subject separately (by fitting each subject's behavioral data to its own logistic function), we discovered that 2/3 of our

**Table 3. Influence of the _value distance_ on the choice proportion of the _target_.**

| | Experiment 1 (n = 38) | | | | Replication (n = 81) | | | |
|---|---|---|---|---|---|---|---|---|
| **Fixed-effects Parameters** | **B** | **SE#** | **Z** | **p-val** | **B** | **SE#** | **Z** | **p-val** |
| Constant | 0.19 | 0.06 | 3.323 | < .001*** | 0.17 | 0.04 | 4.244 | < .001*** |
| _Value distance_ | -0.25 | 0.12 | -2.09 | .03 * | -0.18 | 0.08 | -2.12 | .03 * |
| **Random-effects Parameters** | **var** | | | | **var** | | | |
| Constant | 0.03 | | | | 0.04 | | | |

We used mixed effect logistic regression with random intercept.

# Robust Std. Err. (Errors clustered by Subject)

* p < .05 **p < .01

*** p < .001.

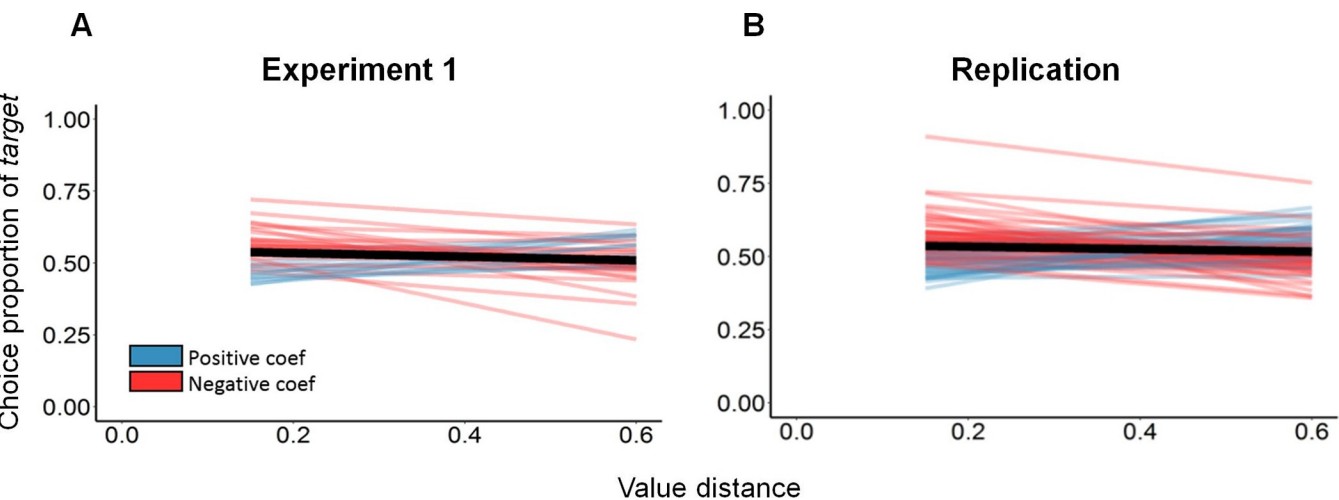

**Fig 5. Choice proportion of the *target* as a function of *value distance*.** Red color represents subjects who had a negative slope (as in the overall coefficient we found in the main regression), while blue color represents subjects who had a positive slope. The black bold line represents the mean slope across all subjects. **(A) Experiment 1:** 68% of subjects had negative slope coefficients, while 32% had positive slope coefficients. **(B) Replication:** 61% of subjects had negative slope coefficients, while 39% had positive slope coefficients.

subjects had negative slope coefficients similar to the overall coefficient we found in the main regression (i.e., a negative influence of the VD on the probability to choose the *target*), while the other 1/3 had positive slope coefficients (i.e., a positive influence of the VD on the probability to choose the target) (Fig 5A).

Additionally, there was no connection between the size of the *attraction effect* of a specific subject (choice proportion of *target*) and the tendency to be affected negatively or positively by the VD ($R = 0.09$, $p = 0.6$).

Since there was such a large variability between subjects, we decided to use a model with a random slope in addition to the random intercept. That is, we allowed the intercept and the slope coefficients of the VD to vary across subjects in addition to clustering the errors per subject. Using this model, we still found a marginally significant effect of the VD on the proportion to choose the *target* option ($\beta = -0.25$, $p = 0.07$; left side of S1 Table). Note, that the coefficient value is the same as in the previous model. Therefore, we concluded for these series of analyses, that on average there is a small negative effect of the *value distance* between the *decoy* and *target* on the proportion to choose the *target* option. However, there is a large variation across subjects in their individual slope coefficients (variance = 0.87).

**Sensitivity to physical proximity influences the *attraction effect*.** Next, we wanted to examine, across subjects, if and to what extent, there is an influence of the sensitivity to physical proximity (as measured in the Gestalt task) on the propensity to demonstrate the *attraction effect*. Therefore, we added the subject-specific Gestalt sensitivity threshold parameter that we estimated in the Gestalt task as another predictor to our model.

Interestingly, as can be seen in the left side of Table 4 and in the simple correlation presented in Fig 6A (for illustration purposes only), the Gestalt sensitivity threshold had a significant negative effect on the proportion to choose the *target* ($\beta = -0.04$, $p<0.03$). That is, the lower the Gestalt sensitivity threshold of a given subject (more sensitive to the proximity law), the more the subject tended to choose the *target* option. Importantly, note that the coefficient size and the significance of the VD regressor did not change after introducing the Gestalt sensitivity regressor, suggesting that the effect of the Gestalt sensitivity on choice is orthogonal to the effect of the VD on choice.

**Table 4. Summary of the mixed effects logistic regression model for variables predicting the choice proportion of the *target*.**

| Fixed-effects Parameters | Experiment 1 (n = 38) | | | | Replication (n = 81) | | | |
|---|---|---|---|---|---|---|---|---|
| | **B** | **SE#** | **Z** | **p-val** | **B** | **SE#** | **Z** | **p-val** |
| Constant | 0.46 | 0.13 | 3.53 | < .001*** | 0.39 | 0.09 | 4.45 | < .001*** |
| *Value distance* | -0.25 | 0.14 | -1.86 | .06 | -0.17 | 0.09 | -1.83 | .07 |
| Gestalt Threshold | -0.04 | 0.02 | -2.26 | .02 * | -0.03 | 0.01 | -2.72 | < .01 ** |
| **Random-effects Parameters** | **var** | | | | **var** | | | |
| Constant | 0.00 | | | | 0.00 | | | |
| *Value distance* | 0.14 | | | | 0.15 | | | |

# Robust Std. Err. (Errors clustered by Subject)

* p < .05

** p < .01

*** p < .001.

To exclude the possibility that the significant negative link between the Gestalt threshold and the probability to choose the *target* is merely due to task engagement, such that subjects who were less engaged in the Gestalt task (and thus have higher thresholds) were also less engaged in the Decoy task (and thus have lower *attraction effect* sizes), we performed further analyses and added them to the supplementary material (S1 Text).

In the perceptual task, in order to examine task engagement, we measured the slope of the logistic regression fit for each subject. The meaning of the slope of the logistic fit is how accurate was the subject in general, across all intervals (distribution of error rates across trial difficulties). We, then, used the Gestalt slope as a predictor in our main analysis instead of the Gestalt threshold, and had no significant effect of the Gestalt slope on the probability to choose the *target* in both experiments (Experiment 1: $\beta = 0.12$, $p = 0.52$; Replication: $\beta = 0.04$, $p = 0.67$; Table 1 in S1 Text). These results indicate that there is no systematic effect of the error rates (task engagement) in the Gestalt task and the level of choice proportion of the *target* in the Decoy task.

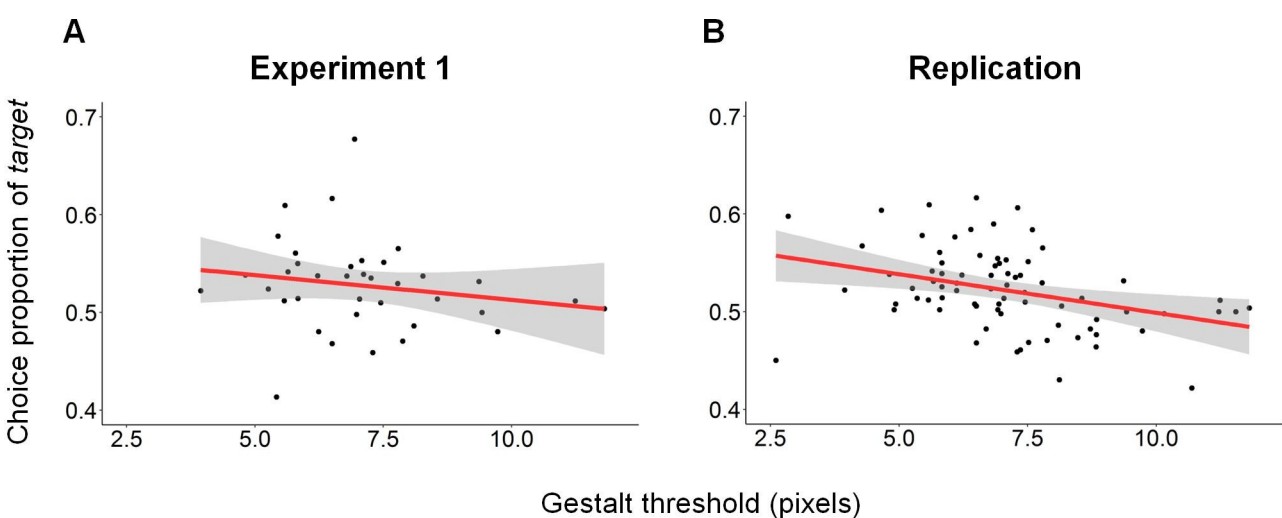

**Fig 6. Correlation between Gestalt threshold and choice proportion of the *target*.** The lower the Gestalt sensitivity threshold of a given subject (more sensitive to the proximity law), the more the subject tends to choose the *target* option. **(A) Experiment 1:** R = -0.31, p = 0.06, n = 38. **(B) Replication:** R = -0.25, p = 0.02, n = 81.

Regarding the Decoy task, it is impossible to define a choice error since there is no correct answer in each trial (except for the first order stochastically dominated trials in which all the subjects, except one, chose the 100% winning probability options all the time). Nonetheless, equivalently to the slopes of the logistic fits in the Gestalt task, we measured the choice variance in each trial type in the Decoy task (there were 32 different trial types that were repeated 8 times each). We calculated two measurements for task engagement in the Decoy task: 1) the mean of choice variance which gives an indication of how consistent was the subject per trial type (the smaller this mean of choice variance, the more consistent was the subject per trial type and thus, more engaged in the Decoy task), and 2) the variability across trial types which represents if the subject responded differently across the different trial types (the smaller the variability of choices across trial types, the less the subject changed his response according to the different trial types, and thus, we assume, the less engaged she was in the task) [detailed equations of the two measurements are available in S1 Text]. When we examined the correlation between each of these measurements of task engagement (the mean of choice variance per trial type and the variability of choices across trial types) and the choice proportion of the *target*, we observed that there is no significant correlation between neither of the measurements for task engagement in the Decoy task and the choice proportion of the *target* (Fig 7 in S1 Text). This indicates that subjects who had a higher variance in their choices per trial type or a small variability across trial types, and thus were probably less engaged in the Decoy task, did not choose systematically the *target* more or less often.

Moreover, there is no significant correlation between neither of the measurements for task engagement in the Decoy task (the mean of choice variance per trial type and the variability across trial types) and the measurement of task engagement for the Gestalt task (the slope of the logistic fit) (Fig 8 in S1 Text) which demonstrates that there is no connection between the levels of task engagement in both tasks.

Finally, we ran a regression analysis which includes VD, Gestalt threshold, and the task engagement measurements (Gestalt slope for the Gestalt task and the mean of choice variance per trial type for the Decoy task) as predictors to the choice proportion of the *target* for both experiments and observed that none of the task engagement measurements had a significant effect on the choice proportion of the *target* in both experiments (Table 2 in S1 Text). Furthermore, the coefficients of our main predictors (Gestalt threshold and VD) of the model which includes the task engagement measurements (Table 2 in S1 Text) were very similar to the coefficients of our main predictors in our main model in the paper (Table 4) in both experiments.

These results suggest that the effect of the sensitivity to the proximity law on the choice proportion of the *target* is not related to task engagement (see S1 Text for more details).

**Range decoys as oppose to frequency decoys induce a stronger *attraction effect*.** It was previously shown that range decoys (which gives the *target* an advantage in its weaker attribute) produce stronger *attraction effects* in comparison to frequency decoys (which gives the *target* an advantage in its stronger attribute) [1,31]. In order to examine if this is the case in our data, we added the decoy type (range or frequency) as a dummy predictor to our model. Similar to the findings of previous studies, range decoys were associated with a higher probability to choose the *target* in comparison with frequency decoys (dummy variable: range was coded as 0. $\beta = -0.12$, $p < 0.01$; left side of S2 Table). Importantly, the size and significance of all other regressors remained the same.

## Replication experiment

Our aim was to replicate the results of Experiment 1. Therefore, we pre-registered the results of experiment 1 and the planned replication experiment (see OSF https://osf.io/jzk6y/), which was identical to Experiment 1, both in design and analysis.

## Subjects

81 valid participants completed the replication experiment (demographic statistics are reported in Table 1). 21 additional subjects were excluded based on our pre-registered exclusion criteria: five performed poorly in the Gestalt task and thirteen performed poorly in the Decoy task. Three performed poorly on both tasks.

## Gestalt results

Similar to Experiment 1, we found that the propensity to discriminate between the two stimuli increased as a function of the interval increase between the dots demonstrating that subjects were very sensitive to the spaces between the dots ($\beta$ = 0.38, p<0.001; right side of Table 2). Furthermore, in the replication experiment, the average sensitivity threshold was 7.02 (±0.2) pixels (ranging from 2.61 to 11.8 pixels) [Fig 3B, bottom histogram] which is very similar to the distribution of sensitivity thresholds in Experiment 1 (t = 0.09, p = 0.92) [Fig 3B, top histogram].

## Decoy results

We present here only the results of the full model of the replication experiment. However, we performed the same analysis steps as was presented in Experiment 1. The partial models of the replication experiment are presented at the right side of Tables 3 and 4 as well as at the right side of S1 and S2 Tables, and are also available online at https://osf.io/jzk6y/.

Similar to Experiment 1, on average, subjects chose the *target* significantly higher than chance level (one sample t-test, mean = 0.53, CI = [0.51, 0.54], t(80) = 4.14, p<0.001). Additionally, there was a high variability across subjects in their probability to choose the target (the range of probabilities spreads between 0.41 and 0.83 (Fig 4B)). Similar to Experiment 1, ~20% of the subjects displayed a significant decoy effect on an individual level (17 out of 81 subjects (21%) chose the target significantly different than 50% (p<0.05); detailed individual results are available in S3 Appendix). Moreover, similarly to Experiment 1, most of the subjects who showed a significant *decoy effect* displayed an *attraction effect*, while 18% displayed a *repulsion effect*.

Interestingly, the coefficients of our predictors of the full model for the replication experiment were very similar to the coefficients of the full model for Experiment 1 (right side of Table 4; and of S2 Table). Specifically, we again found a marginally negative effect of the VD on the proportion to choose the *target* option ($\beta$ = -0.17, p = 0.07; right side of Table 4). Moreover, we observed a similar proportion of negative and positive coefficients as was found in Experiment 1 (**Experiment 1:** 68% of subjects had negative coefficients (Fig 5A); **Replication:** 61% of subjects had negative coefficients (Fig 5B)).

Regarding the connection between the Gestalt sensitivity threshold and the probability to choose the *target* in the Decoy task, we again observed a significant negative effect of the Gestalt threshold on the proportion to choose the *target* ($\beta$ = -0.03, p<0.01; right side of Table 4). These results strengthen our conclusion from Experiment 1 that the lower the Gestalt sensitivity threshold of a given subject (more sensitive to the proximity law), the more the subject tends to choose the *target* option.

Furthermore, similar to Experiment 1, range decoys were associated with a higher probability to choose the *target* in comparison with frequency decoys (dummy variable: range was coded as 0. $\beta$ = -0.06, p<0.05; right side of S2 Table).

## General discussion

In the current study, we aimed to elucidate the mechanisms underlying the *attraction effect* by examining a potential link between this effect and a well-known perceptual phenomenon, the

Gestalt law of proximity. Across two independent and identical experiments with a pre-registered replication, we found that the lower the Gestalt sensitivity threshold of a given subject as measured in a perceptual task (i.e., it is easier for this subject to differentiate between the two stimuli because her tendency to group by the Gestalt law of proximity is higher), the more she tends to choose the *target* option (i.e., displays a stronger *attraction effect*). Therefore, we suggest that the variation across subjects in their susceptibility to the Gestalt law of proximity might account for some of the variation observed in their tendency to show the *attraction effect*. These results strengthen the notion that there are commonalities between perceptual and value-based processing by demonstrating a within-subject link between a perceptual phenomenon (proximity law) and a value-based bias (*attraction effec*t). Moreover, our findings can help us better understand the mechanisms underlying the *attraction effect* using the within-subject link with the Gestalt law of proximity.

### Grouping by proximity as an optional mechanism for the *attraction effect*

How can grouping by proximity be one of the mechanisms mediating the *attraction effect*? The Gestalt principles of self-organization aim to describe how our brain engineers the perception of the world around us [32]. Since we live in a noisy world with endless sensory information but at the same time with constraints of limited resources and capacity, we are naturally in a need for an efficient coding [41,42] in order to balance between robustness (stability, solve ambiguity and resistance to change) and flexibility (dynamic environment). The Gestalt principles offer a solution for this computational problem of balancing robustness and flexibility using both the internal and external aspects of perception [43].

The perceptual system, which is established to be subordinate to specific rules of efficient processing, is an integral input into the value-based decision process [44]. Therefore, it is not surprising that both sensory and choice biases are considered to be the consequence of several canonical computations and patterns in the brain, which we use in order to efficiently code our environment [26,45–47]. Our results demonstrate a direct link between these two domains. The within-subject relationship between the sensitivity to group by proximity and the susceptibility to the *attraction effect* suggests that the grouping principle of proximity (either physical or value-based proximity) might be a part of the mechanism underlying the *attraction effect*. We offer a theoretical model of how this connection might occur using attention as a mediator (Fig 7).

There is a close interplay between selective attention and perceptual organization processes [48,49]. Since, as we mentioned above, we have limited resources and capacity [41,42], we are obliged to direct our attention to a particular part of the scene [50]. Several studies demonstrated that perceptual organization plays a crucial role in the deployment of attention [48,51,52]. For example, Kimchi and colleagues [51] demonstrated that stimuli which are grouped according to principles of self-organization attract subjects' attention more than stimuli which are not perceived as grouped. Therefore, selective attention might be a mediator

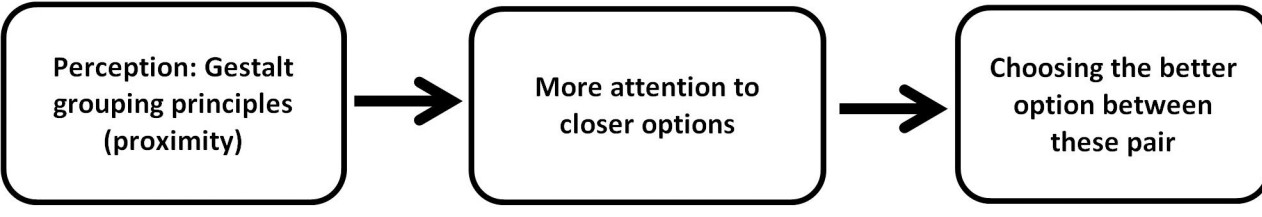

**Fig 7. A suggested mechanism of how grouping by proximity may mediate the *attraction effect*.**

which allows Gestalt principles of self-organization to increase the efficiency of our perception process.

Interestingly, selective attention has also been suggested to have an integral role in the mechanism underlying the *attraction effect*. For instance, Roe and colleagues [16] posit in their Multialternative Decision Field Theory (MDFT), that the shifting of attention between the attributes of the choice options which are more similar, increases the attractiveness of the better option between them via lateral inhibition, and leads to the *attraction effect*. Another, more recent model, Multiattribute Linear Ballistic Accumulator (MLBA) model [17] also suggests that more attention is allocated to the comparison between the *target* and the *decoy* options because it is harder to discriminate between them (more similar), and this highlights the superiority of the *target*. The Multialternative Decision by Sampling (MDbS) model [14] also postulates that similar options attract more attention, but because it is easier (rather than harder) to compare between them. Moreover, a recent study demonstrated that the value of the choice options also influence the allocation of attention [53]. Since selective attention plays an integral role both in the Gestalt principles of self-organization and in the *attraction effect*, we propose that the within-subject correlation that we found between sensitivity to proximity and the tendency to show the *attraction effect* could be mediated by selective attention (Fig 7).

A first step of every evaluation or choice process is a perceptual appraisal of the alternatives [44] which, as was mentioned above, is subordinate to specific rules of efficient processing, for example grouping by proximity. Therefore, in the Decoy task, we propose that when the subject is presented with 3 different gambling options, the Gestalt self-organization principles will construct the way she will perceive these options. Since, some of the options are closer to each other in the value space, she will tend to group these options according to the proximity law. Moreover, she will direct her attention to these closer options (in the value space), because they are perceived as more similar, and according to the self-organization rules, closer (similar) options receive more attention. This in turn, will lead to more comparisons between the closer options and thus to the *attraction effect*. According to the MDbS [14] and MDFT [16] models, the higher the probability of comparison between a specific pair, the higher the probability to choose the better option between that pair, which is the definition of the *attraction effect*.

An unresolved question is what actually leads people to drive more attention to similar options. We suggest the Gestalt principle of grouping by proximity as a possible explanation for this query, since we showed that subjects who are less sensitive to grouping by proximity are also less susceptible to the *attraction effect*. Furthermore, when the *decoy* was located further away from the *target*, subjects tended to display a weaker *attraction effect*. However, note, that this is a theoretical notion that should be examined in future studies using either imaging techniques or eye movements.

## The effect of *value distance* on the *attraction effect*—heterogeneity between subjects

We replicated the *attraction effect* when averaging the behavior across subjects. The probability of choosing the *target* was significantly higher when the *decoy* was asymmetrically dominated by the *target* than when it was asymmetrically dominated by the *competitor*. However, our results demonstrated a considerable heterogeneity between subjects in their sensitivity to the *attraction effect*. We observed, in both experiments, that only ~20% of the subjects displayed a significant *decoy effect* on an individual level. It is important to note that in most previous studies, only group effects were described [1–3,14], either because the study was a between-subject's design or because the study only focused on group effects. However, studies that did

examine and report results at the individual level show that there are systematic differences across subjects in regard to the influence of context on their behavior [40,54,55] and posit that *decoy effects* are usually weak effects [40,54] similar to our results.

Furthermore, although we aimed to reach for each subject an indifference between options A and B using the Calibration task, the safer option (option A) was chosen more often across subjects in the Basic condition in both experiments, albeit only significant in the Replication experiment. Additionally, for around third of the subjects in both experiments there was a significant difference in the subjective value between the two options even though we used a Calibration task. This could be another reason for the small *attraction effect* sizes in our study. Nonetheless, although the calibration task did not work perfectly, we were able to show a significant *attraction effect* across subjects as well as a significant link between the choice proportion of the *target* and the susceptibility to group by proximity.

Moreover, across both experiments, we demonstrated a marginally significant negative VD effect on the *attraction effect*. That is, the further away the *decoy* was from the *target*, the less the subject chose the *target*, and hence, the lower was the *attraction effect*. Interestingly, this is in line with the manner physical distance between objects affects the susceptibility to grouping by the proximity law; the bigger the physical distance between objects, the less chance there is to perceive these objects as grouped by proximity [32]. The effect is only marginally significant because there is a considerable variability in subjects' sensitivity to the VD. Two thirds of our subjects, had negative slope coefficients similar to the overall coefficient we found in the main regression (i.e., a negative influence of the VD on the probability to choose the *target*), while the other third had positive slope coefficients (i.e., a positive influence of the VD on the probability to choose the *target*).

Additionally, there is evidence for both negative and positive effects of the *target-decoy* distance on the *attraction effect*. Several previous studies found that further *decoys* produced a smaller *attraction effect*, similar to our findings [33,40]. However, other studies observed the opposite effect. For example, Soltani et al. [4] found that close *decoys* had no significant effect while far *decoys* had a very strong effect, and Spektor et al. [38] demonstrated that an increase in the *target-decoy* distance of perceptual stimuli increased the choice proportion of the *target*. A possible explanation for this discrepancy and the large variability between subjects in our study is that there are actually two contradicting forces in the Decoy task: on the one hand, the more similar the *decoy* and the *target* are (smaller VD), the more attention subjects would allocate to these options which would lead to a more frequent comparison between them (which would then result in a larger *attraction effect*) [14]. On the other hand, the more the *decoy* is inferior to the *target* (larger VD), the more the subjects would perceive the superiority of the *target* (larger *attraction effect*) [see also [56]]. In fact, both MDFT model [16] and MDbS model [14] refer to the point that when the *decoy* is very similar to the *target* and hence its inferiority is less clear, it may reduce the *attraction effect*. Therefore, we suggest this balance between the similarity of the *decoy* and the *target*, and the inferiority of the *decoy* in comparison to the *target*, as a possible explanation for the contradicting findings in the literature regarding the effect of the *target-decoy* distance on the *attraction effect*, and for the large variability between subjects in the effect of VD on the *attraction effect* in our study. It might be that these subjects who displayed a positive effect of the VD on the probability to choose the *target* are more sensitive to the inferiority of the *decoy*, while the subjects who displayed a negative effect are more sensitive to the similarity between the *target* and the *decoy*. It is also important to note that in our study the smallest *target-decoy* distance was a difference of 15% while in the other contradicting findings the smallest *target-decoy* distance was 2% [4,38]. These different ranges may also interact with the two contradicting forces of similarity and inferiority which affect the size of the *attraction effect*. A difference of 2% between the *decoy*

and the *target* may result in a *decoy* which is not inferior enough to the *target*, and thus subjects would display less *attraction effect*, while a difference of 15% may be large enough for the *target* to be perceived as superior to the *decoy* but still similar enough to it.

Nevertheless, our results suggest that some of the variability between subjects in their overall susceptibility to the *attraction effect* can be explained by their sensitivity to the Gestalt law of proximity. Subjects who displayed low sensitivity to the *attraction effect* or even displayed the opposite effect–*repulsion effect*, were also less susceptible to grouping by proximity. However, our findings also highlight the importance of examining variability across subjects and not relying only on group differences in order to understand behavior and cognition.

## Conclusion

Our findings provide evidence for a within-subject link between the sensitivity to a perceptual heuristic (proximity law of Gestalt theory) and the sensitivity to a value-based bias (*attraction effect*). These findings elucidate the commonalities between sensory and value-based processing within an individual. This also strengthens the notion that the brain generalizes across domains. Specifically, we suggest that the variation across subjects in their susceptibility to the Gestalt law of proximity might account for some of the variation observed in the *attraction effect*. Therefore, we used the comprehensive research and knowledge regarding the proximity law of Gestalt theory in order to explain a query in the mechanism underlying the *attraction effect*. Previous studies suggested that selective attention to more similar options plays an integral role in the mechanism underlying the *attraction effect* [14,16,17]. However, an unresolved question is what actually leads people to drive more attention towards similar options. Using the evidence that there is a close interplay between selective attention and the Gestalt grouping principles, we suggest that grouping by proximity of the more similar options is what leads people to drive more attention to these similar options. This allows us to draw a specific connection between perceptual processing (grouping by proximity) and value-based processing (comparison between lottery options). These findings are important to better understand the mechanisms underlying the *attraction effect*. Future work could examine computational models that may suggest further explanation for the mechanism underlying this interesting connection between the proximity law and the *attraction effect*.

Furthermore, our results offer a new approach for examining mechanisms of context-based choice biases using perceptual mechanisms. We can use the evidence that the brain generalizes across domains and that there are fundamental rules that it follows, in order to transfer knowledge from one domain to the other. In addition, finding such connection between perceptual and value processing may shed light on the overall mechanism by which the brain integrates information across different domains.

## Supporting information

**S1 Table. Influence of the *value distance* on the choice proportion of the target.**
(PDF)

**S2 Table. Summary of the mixed effects logistic regression model for variable predicting the choice proportion of the *target*.**
(PDF)

**S1 Appendix. List of ssstrials for the calibration task.**
(PDF)

**S2 Appendix. Calculation of the decoy options.**
(PDF)

**S3 Appendix. Individual results of binomial tests for choice proportion of *target*.**
(PDF)

**S1 Text. Additional analyses.**
(PDF)

## Acknowledgments

We thank Vered Kurtz-David, Adam Hakim, Tal Sela, Sharon Yefet and Noa Palmon for their help and guidance in many discussions. We also thank Marius Usher for his valuable advice throughout the work on the study.

## Author Contributions

**Conceptualization:** Liz Izakson, Dino J. Levy.

**Data curation:** Yoav Zeevi.

**Formal analysis:** Liz Izakson, Yoav Zeevi.

**Funding acquisition:** Dino J. Levy.

**Investigation:** Liz Izakson, Yoav Zeevi, Dino J. Levy.

**Methodology:** Liz Izakson.

**Project administration:** Liz Izakson.

**Software:** Liz Izakson.

**Supervision:** Dino J. Levy.

**Validation:** Liz Izakson.

**Visualization:** Liz Izakson.

**Writing – original draft:** Liz Izakson.

**Writing – review & editing:** Liz Izakson, Dino J. Levy.

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
