## [Decision Letter · Decision Letter 0]

30 Jun 2020

PONE-D-20-14306

Attraction to similar options: the Gestalt law of proximity is related to the attraction effect

PLOS ONE

Dear Dr. Izakson,

Thank you for submitting your manuscript to PLOS ONE. After careful consideration, we feel that it has merit but does not fully meet PLOS ONE’s publication criteria as it currently stands. Therefore, we invite you to submit a revised version of the manuscript that addresses the points raised during the review process.

Two expert reviewers have reviewed the submission and are in agreement that the study described is a useful contribution to the field, pending some conceptual and methodological revisions. I agree with their reviews and thus will not reiterate them here. The recommendations from both reviewers are straightforward and concrete, and thus should be addressable in a major revision. Please make sure to address each point in your resubmission.

We look forward to receiving your revised manuscript.

Kind regards,

Tyler Davis, Ph.D.

Academic Editor

PLOS ONE

Journal Requirements:

Reviewers' comments:

Reviewer's Responses to Questions

**Comments to the Author**

1. Is the manuscript technically sound, and do the data support the conclusions?

Reviewer #1: No

Reviewer #2: Partly

2. Has the statistical analysis been performed appropriately and rigorously? 

Reviewer #1: Yes

Reviewer #2: Yes

3. Have the authors made all data underlying the findings in their manuscript fully available?

Reviewer #1: Yes

Reviewer #2: Yes

4. Is the manuscript presented in an intelligible fashion and written in standard English?

Reviewer #1: Yes

Reviewer #2: Yes

5. Review Comments to the Author

Reviewer #1: In their paper, the authors tested whether "physical proximity" and "value proximity" are related concepts. To do so, they investigated the relationship between the attraction effect, a phenomenon according to which the addition of a specific inferior option affects the relative choice proportions between the original options, and the Gestalt law of proximity, the tendency to group objects according to their physical proximity. The authors found a correlation between the tendency to group stimuli by proximity and the degree to which they showed an attraction effect.

The paper investigates an interesting hypothesis and extends our understanding of the similarities between perceptual and value-based decision making. The paper is well written and structured, the methods are described in detail, and the analyses are conducted rigorously. In general, I think this is a good paper worthy of publication. However, I have a few minor questions and one major concern regarding the theoretical background and the interpretation of the results that, in my opinion, has to be addressed prior to publication.

Major comment:

1) The authors motivate their research by the question whether the perception of value distance is similar (or analogous) to the perception of actual physical distance. They use a psychophysical task in which individuals have to tell whether two arrays of dots are identical or different to determine the sensitivity threshold and a standard risky-choice task to determine the degree to which individuals exhibit the attraction effect. A negative correlation between the threshold (i.e., higher values = lower sensitivity) and the attraction effect (i.e., higher values = stronger influence of context) emerges which the authors interpret as a commonality, even claiming that the results "strengthen the notion that the brain generalizes across domains" (without it being a neuroscientific investigation).

In the perceptual task, individuals faced a total of 192 trials, out of which 174 were between two different arrays and 18 were between two identical arrays. Despite an achievable accuracy of more than 90% by merely pressing "different" on every trials (individuals that the authors excluded), there have been substantial individual differences with respect to sensitivity in the task. The authors interpreted that those with a low threshold are more sensitive to a perceptual heuristic. I find this classification quite surprising, given that individuals with a low threshold are those that performed the task more accurately (objectively better) than those with a high threshold. To me, it seems that those with a higher threshold are simply less engaged with the task/more random in their response behavior. Strikingly, random behavior would also be reflected in a regression toward 50% choices of the target in the decoy task (i.e., weaker attraction effect).

If this is the case, the observed correlation is merely a statistical necessity reflecting randomness due to boredom/task engagement, such that more random people have both a higher threshold and a smaller attraction effect. The conclusions of the authors would not hold in such a scenario. Thankfully, the authors' design of both task allows to distinguish the two interpretations: In both tasks, each unique trials is repeated (at least) 8 times. This property can be used to quantify the individuals' consistency, in other words, the proportion of choosing the same option across repetitions. A person that is engaged in the task but is simply not very sensitive will have a high consistency and a rather abrupt cut-off as the variable stimulus becomes less similar to the constant one. In contrast, a person that is less engaged will have a much more uniform distribution of error rates across the different difficulties. The analogous reasoning applies in the risky choice task.

The authors should provide a more convincing reasoning why individuals with a lower threshold are more susceptible to a heuristic and present analyses that rule out that the confound of task engagement is the main driver of the observed effect.

Additional minor comments:

1) The authors mention on multiple occasions that there are substantial individual differences and that about a quarter of their sample show the opposite of the attraction effect, the repulsion effect. The presence of individuals on the "other" side of the effect is more-or-less a necessity; If these individuals were not there, the observed effect size would be substantially greater. If the authors want to make this claim, they would need to run tests on an individual level (e.g., a binomial test). Most likely, only few of those people (if any) will show a significant deviation from a 50% chance of choosing the target option.

2) As a robustness check, the authors checked for violations of WASRP. A correlation of r=.99 suggests that the main analysis and the supplemental analysis are almost identical. However, I do not see how these analyses differ from each other. Does either of the analyses include trials on which the decoy was chosen? It would be helpful if the authors could clarify the differences between the analyses.

3) In the basic condition, option A seems to be preferred to option B (60% vs. 40%, Fig. 5 in S3), at least on average. Since this is not mentioned anywhere in the text I have to guess, but I assume that A is the safe option and B is the riskier option. Moreover, there are substantial differences in the choice proportions, such that the values range from 0% to about 80%, whereas they should be closer to 50%. The authors should at least mention and briefly discuss that their calibration procedure did not achieve the desired result.

Reviewer #2: This is my first review of the paper “Attraction to similar options: the Gestalt law of proximity is related to the attraction effect” by Izakson et al.. In their study, the authors investigate if and how common processes underlying perceptual and value-based decisions might cause the attraction effect. In two studies, one of them a pre-registered replication study, they basically found a correlation between susceptibility to the Gestalt law of proximity and the size of the attraction effect. While this is an interesting finding that should be published, the authors thus far did not aim at understanding why this correlation exists. I therefore recommend to do additional analyses. I furthermore feel uncomfortable with the number of excluded participants and the reasons for exclusion. Please find my details comments below.

1.) Please define spatial context.

2.) I would not call attraction effect, compromise effect and similarity effect “decoy effects” but “context effects”.

3.) Line 114: Transformation into subjective scale: please use a “weaker” formulation. At least for value this notion is based on models and not on evidence. Other models take objective values as inputs without transforming them into a subjective scale.

4.) “Contemporary decision making theory” is a very unusual term in decision science.

5.) More than 20% of the participants were excluded, which obviously influences the results. However, I do not see obvious “mistakes” resulting in exclusion but relevant behavior. Some people might not be able to perform well in the Gestalt task. Yes, if they chose “different” in identical trials they might have had a prior. But many other participants might have had, too. Perhaps they chose different when the distance was 20.5 pixels just because they had a prior and not because they perceive a difference. I do not see the 50% threshold as a valid exclusion criterion. Similarly, the slope of the logistic regression. Obviously, these people exist. Why should their behavior not map to behavior in the decoy task? How was the “more than 96% criterion chosen”? Also these participants not necessarily make mistakes but show their preferences. I would wish to see analyses including all participants.

6.) Criterion for attraction effect is >50% choice probability for target. Given some noise, I would always expect some individuals above or below 50%, which is not necessarily a sign for the attraction or repulsion effect.

7.) Logistic regressions: Instead of focusing on significance levels of slopes, I would want to see a model comparison of models including value difference or not (e.g. based on BIC).

8.) If I understand the analyses right, the main finding is a correlation between susceptibility to the Gestalt law of proximity and the size of the attraction effect. The question that now arises is why. I would encourage the authors to aim at answering this question. At the moment two behavioral outcome variables are correlated. An interesting approach would be to identify models that predict these behavioral outcomes variables (e.g., MDFT) and see if parameters of these models are correlated. Ideally a single model can be defined predicting both behavior in the Gestalt and the decoy task, including a single parameter driving observed correlations between behavioral outcome variables.

6. PLOS authors have the option to publish the peer review history of their article (what does this mean?). If published, this will include your full peer review and any attached files.

Reviewer #1: **Yes: **Mikhail S. Spektor

Reviewer #2: No

---

## [Author Response · Author response to Decision Letter 0]

6 Sep 2020

Re: PONE-D-20-14306

Attraction to similar options: the Gestalt law of proximity is related to the attraction effect

Dear Editor,

We sincerely thank you and the reviewers, for the time and effort taken to help us improve our manuscript. We hope that we adequately addressed the concerns raised by the reviewers. Please find below the comments we received in bold, and our responses in italics. Note that all the additions and corrections in the main text and the supplementary information are highlighted in yellow.

Concerns raised by Reviewer 1:

Major comments 

The paper investigates an interesting hypothesis and extends our understanding of the similarities between perceptual and value-based decision making. The paper is well written and structured, the methods are described in detail, and the analyses are conducted rigorously. In general, I think this is a good paper worthy of publication. However, I have a few minor questions and one major concern regarding the theoretical background and the interpretation of the results that, in my opinion, has to be addressed prior to publication.

 The authors motivate their research by the question whether the perception of value distance is similar (or analogous) to the perception of actual physical distance. They use a psychophysical task in which individuals have to tell whether two arrays of dots are identical or different to determine the sensitivity threshold and a standard risky-choice task to determine the degree to which individuals exhibit the attraction effect. A negative correlation between the threshold (i.e., higher values = lower sensitivity) and the attraction effect (i.e., higher values = stronger influence of context) emerges which the authors interpret as a commonality, even claiming that the results "strengthen the notion that the brain generalizes across domains" (without it being a neuroscientific investigation).

In the perceptual task, individuals faced a total of 192 trials, out of which 174 were between two different arrays and 18 were between two identical arrays. Despite an achievable accuracy of more than 90% by merely pressing "different" on every trials (individuals that the authors excluded), there have been substantial individual differences with respect to sensitivity in the task. The authors interpreted that those with a low threshold are more sensitive to a perceptual heuristic. I find this classification quite surprising, given that individuals with a low threshold are those that performed the task more accurately (objectively better) than those with a high threshold. To me, it seems that those with a higher threshold are simply less engaged with the task/more random in their response behavior. Strikingly, random behavior would also be reflected in a regression toward 50% choices of the target in the decoy task (i.e., weaker attraction effect).

If this is the case, the observed correlation is merely a statistical necessity reflecting randomness due to boredom/task engagement, such that more random people have both a higher threshold and a smaller attraction effect. The conclusions of the authors would not hold in such a scenario. Thankfully, the authors' design of both task allows to distinguish the two interpretations: In both tasks, each unique trials is repeated (at least) 8 times. This property can be used to quantify the individuals' consistency, in other words, the proportion of choosing the same option across repetitions. A person that is engaged in the task but is simply not very sensitive will have a high consistency and a rather abrupt cut-off as the variable stimulus becomes less similar to the constant one. In contrast, a person that is less engaged will have a much more uniform distribution of error rates across the different difficulties. The analogous reasoning applies in the risky choice task.

The authors should provide a more convincing reasoning why individuals with a lower threshold are more susceptible to a heuristic and present analyses that rule out that the confound of task engagement is the main driver of the observed effect.

Response:

We thank the reviewer for the kind words, and think that these are very important points regarding the relation between heuristic and accuracy and the potential confound of task engagement. We agree that it might seem odd that subjects with a lower threshold are more sensitive to the heuristic. However, this is a matter of the definition of a heuristic. In the last decade, there is substantial literature regarding efficient coding which treat biases and heuristics, both sensory and value-based, as optimal with respect to the environmental conditions (Polania et al., nature neuroscience, 2019; Louie et al., Ann N Y Acad Sci., 2012; Wagemans et al., Psychol bull, 2012). That is, in certain situations, the heuristic can be considered as efficient under the relevant conditions and constraints, and hence should not always be treated as an error. Moreover, our “task was based on a psychophysical task presented in (Gori et al., Vision Research, 2010) where a higher tendency to detect differences in physical distance as well as the tendency to group by proximity is translated to a lower threshold. A subject who is more susceptible to grouping by proximity will detect the differences between the constant stimulus and the variable stimulus at a much lower distance between the triplets of dots, since it would be easier for her to group the row of 12 dots into 4 groups of triplets.” We added the part with the quotation marks as a clarification of this point in the description of the Gestalt task in the main text (p. 10, line 248).

Regarding the reviewer’s concern that the level of task engagement might be a confound in the study, we performed the analyses suggested by the reviewer, which yielded that the negative link between the Gestalt threshold and the choice proportion of the target is not merely due to task engagement. We included a general explanation regarding these analyses in the main text under the results section (page 20, line 481):

“To exclude the possibility that the significant negative link between the Gestalt threshold and the probability to choose the target is merely due to task engagement, such that subjects who were less engaged in the Gestalt task (and thus have higher thresholds) were also less engaged in the Decoy task (and thus have lower attraction effect sizes), we performed further analyses and added them to the supplementary material (S3 text). 

In the perceptual task, in order to examine task engagement, we measured the slope of the logistic regression fit for each subject. The meaning of the slope of the logistic fit is how accurate was the subject in general, across all intervals (distribution of error rates across trial difficulties). We, then, used the Gestalt slope as a predictor in our main analysis instead of the Gestalt threshold, and had no significant effect of the Gestalt slope on the probability to choose the target in both experiments (Experiment 1: β=0.12, p=0.52; Replication: β=0.04, p=0.67; Table 1 in S3 text). These results indicate that there is no systematic effect of the error rates (task engagement) in the Gestalt task and the level of choice proportion of the target in the Decoy task.

Regarding the Decoy task, it is impossible to define a choice error since there is no correct answer in each trial (except for the first order stochastically dominated trials in which all the subjects, except one, chose the 100% winning probability options all the time). Nonetheless, equivalently to the slopes of the logistic fits in the Gestalt task, we measured the choice variance in each trial type in the Decoy task (there were 32 different trial types that were repeated 8 times each). We calculated two measurements for task engagement in the Decoy task: 1) the mean of choice variance which gives an indication of how consistent was the subject per trial type (the smaller this mean of choice variance, the more consistent was the subject per trial type and thus, more engaged in the Decoy task), and 2) the variability across trial types which represents if the subject responded differently across the different trial types (the smaller the variability of choices across trial types, the less the subject changed his response according to the different trial types, and thus, we assume, the less engaged she was in the task) [detailed equations of the two measurements are available in S3 text]. When we examined the correlation between each of these measurements of task engagement (the mean of choice variance per trial type and the variability of choices across trial types) and the choice proportion of the target, we observed that there is no significant correlation between neither of the measurements for task engagement in the Decoy task and the choice proportion of the target (Fig. 7 in S3 text). This indicates that subjects who had a higher variance in their choices per trial type or a small variability across trial types, and thus were probably less engaged in the Decoy task, did not choose systematically the target more or less often. 

Moreover, there is no significant correlation between neither of the measurements for task engagement in the Decoy task (the mean of choice variance per trial type and the variability across trial types) and the measurement of task engagement for the Gestalt task (the slope of the logistic fit) (Fig. 8 in S3 text) which demonstrates that there is no connection between the levels of task engagement in both tasks.

Finally, we ran a regression analysis which includes VD, Gestalt threshold, and the task engagement measurements (Gestalt slope for the Gestalt task and the mean of choice variance per trial type for the Decoy task) as predictors to the choice proportion of the target for both experiments and observed that none of the task engagement measurements had a significant effect on the choice proportion of the target in both experiments (Table 2 in S3 text). Furthermore, the coefficients of our main predictors (Gestalt threshold and VD) of the model which includes the task engagement measurements (Table 2 in S3 text) were very similar to the coefficients of our main predictors in our main model in the paper (Table 4) in both experiments.

These results suggest that the effect of the sensitivity to the proximity law on the choice proportion of the target is not related to task engagement (see S3 text for more details).”

Additionally, we included a detailed analyses in the supplementary material (S3 text under ‘confound analyses’ section). We added all the following parts with the quotation marks to the supplementary text. The parts that are not with quotation marks are further explanations for the reviewer. 

“We performed further analyses in order to exclude the possibility that the significant negative link between the Gestalt threshold and the probability to choose the target is merely due to task engagement, such that subjects who were less engaged in the Gestalt task (and thus have higher thresholds) were also less engaged in the Decoy task (and thus have lower attraction effect sizes).”

Regarding the perceptual task, we agree with the reviewer that since there were 192 trials, out of which 174 were between two different arrays, and 18 were between two identical arrays, responding “different” all the time would result in a 90% overall accuracy even in cases where a subject is not engaged at all in the task (and as was also mentioned by the reviewer we excluded these subjects and expand more about this in the answer for comment 5 of Reviewer 2). However, as the reviewer suggested, we examined the distribution of choice consistency (variance) across all the different trial difficulties (different pixel interval increases) for each non-excluded subject (see Appendix A at the end of this letter), and also averaged this across subjects (figure below). As shown in the figure below (and as can be seen for each subject in Appendix A), the subjects’ choice variance changed according to the difficulty of the trial: the subjects were more consistent in the easier trials (when the distance between the triplets of dots was either very small or very high) and were more variant in the harder trials (around the area of the threshold (5-8 pixels)). This indicates that the subjects were affected by the trial difficulty and were not simply pressing “different” all the time.

Since the slope of the logistic regression fit is a well-known measurement for task engagement in psychophysical tasks, we examined if there is a relation between it and the choice consistency. In order to perform this analysis, we averaged, for each subject, the choice consistency across all trials, and correlated it with the slope of the logistic regression. We found a very high correlation between the mean of choice variance and the slope of the logistic fit (Experiment 1: R=-0.96, p<0.001, n=38; Replication: R=-0.92, p<0.001, n=81; figure below). This demonstrates that the mean of choice variance of a subject is very similar to the slope of the logistic regression fit.

“The meaning of the slope of the logistic fit is how accurate was the subject in general, across all intervals (distribution of error rates across trial difficulties). That is, a flat curve means a fully random subject, while the steeper the slope, the more accurate the subject is, with a step function being perfectly accurate.” As shown in the correlation we conducted, the lower the slope (flatter slope), the more variable the subject is across the different trial difficulties (higher mean choice variance). This suggests that the higher the error rates (lower slope) and the more variable the subject is across all trials, the less engaged she is in the task. 

Since, the slope of the logistic regression fit is a well-known measurement for task engagement in psychophysical tasks and that there is a significant correlation between the slope and the mean of the choice variance, we will use the slope as our measurement for task engagement in the Gestalt task throughout this discussion and in the supplementary text. “Importantly, in our main analysis (Tables 2-4 in the main text; figures 3 and 6 in the main text), as a first step, we excluded subjects with a slope that was not significantly (p<0.05) higher than 0, meaning that subjects with a uniform distribution of errors across the different difficulties (very low slope) were excluded from the analyses because they were not engaged in the task” (we expand on this point in the answer for comment 5 of Reviewer 2 as well). 

“Moreover, when we used the Gestalt slope as a predictor in our main analysis instead of the Gestalt threshold, we had no significant effect of the Gestalt slope on the probability to choose the target in both experiments (Experiment 1: β=0.12, p=0.52; Replication: β=0.04, p=0.67; Table 1). These results indicate that there is no systematic effect of the error rates (task engagement) in the Gestalt task and the level of choice proportion of the target in the Decoy task. Hence, this strengthens the notion that task engagement was not the driver of the main relationship we found between the two tasks.

Table 1. Summary of the mixed effects logistic regression model for variables predicting the choice proportion of the target. 

 Experiment 1 (n = 38) Replication (n = 81)

Fixed-effect Parameters B SE# Z p-val B SE# Z p-val

Constant 0.15 0.07 1.93 .05 0.15 0.05 3.12 <.01**

Value distance -0.25 0.14 -1.83 .06 -0.17 0.09 -1.80 .07

Gestalt Slope 0.12 0.17 0.64 .52 0.04 0.09 -2.72 .67

Random-effects Parameters var var

Constant 0.00 0.00 

Value distance 0.14 0.16 

# Robust Std. Err. (Errors clustered by Subject); * p<.05 **p<.01 *** p<.001”

Additionally, we ran the same regression using the mean of choice variance in the Gestalt task instead of the Gestalt threshold in order to further validate our conclusion. As presented in table 2, we had no significant effect of the mean of choice variance in the Gestalt task on the choice proportion of the target in both experiments (Experiment 1: β=-0.94, p=0.19; Replication: β=-0.48, p=0.26; Table 2).

Table 2. Summary of the mixed effects logistic regression model for variables predicting the choice proportion of the target. 

 Experiment 1 (n = 38) Replication (n = 81)

Fixed-effect Parameters B SE# Z p-val B SE# Z p-val

Constant 0.56 0.17 3.32 <.01** 0.25 0.08 3..23 <.01**

Value distance -0.25 0.14 -1.81 .07 -0.17 0.09 -1.81 .07

Gestalt mean of choice variance -0.94 0.71 -1.31 .19 -0.48 0.43 -1.12 .26

Random-effects Parameters var var

Constant 0.00 0.00 

Value distance 0.17 0.17 

# Robust Std. Err. (Errors clustered by Subject); * p<.05 **p<.01 *** p<.001

“Regarding the Decoy task, it is impossible to define a choice error since there is no correct answer in each trial (except for the first order stochastically dominated trials in which all the subjects, except one, chose the 100% winning probability options all the time).” Therefore, similarly to what the reviewer suggested, we defined subjects whose behavior (choice) did not change across the different trials (different levels of decoy) as not engaged in the task, and this was our exclusion criteria for the Decoy task (we expand on this point in the answer for comment 5 of Reviewer 2 as well). 

“Nonetheless,” we performed the analysis which the reviewer suggested and, “equivalently to the slopes of the logistic fits in the Gestalt task, we measured the choice variance in each trial type in the Decoy task. There were 32 different trial types (2 different data sets, 2 attributes (probability/amount), 2 different types of decoy (range/frequency), 4 options of VD), which were repeated 8 times each. On the one hand,” as Reviewer 1 pointed out, “we expect that a subject who is engaged in the task, will have a relatively small variance in her choices across repetitions of the exact same trial. On the other hand, analogous to the Gestalt task, we would also expect that a subject who is engaged in the task will have some variance across the different trial types (different levels of decoy). 

Therefore, we calculated two measurements” to address the reviewer’s concern: “1) the mean of choice variance, where we calculated the variance in choice of the 8 repetitions for each trial type, and then averaged for each subject these variances across the 32 trial types (equation 1). This average of choice variance gives an indication of how consistent was the subject per trial type (the smaller this average of choice variance, the more consistent was the subject per trial type and thus, more engaged in the Decoy task), and 2) the variability across trial types (variance of means), where we calculated the probability to choose a specific option (A/B) per trial type and then calculated the variability of choice probabilities across trial types (equation 2). This measurement represents if the subject responded differently across the different trial types (the smaller the variability of choices across trial types, the less the subject changed his response according to the different trial types, and thus, we assume, the less engaged she was in the task). 

Equation 0:

(∑_1^Ni▒x_ijk )/N_i =P_jk

P_jk is the choice probability of option B^* across the 8 repetitions of a trial type, where i stands for repetition (8 per trial type), j stands for trial type (32 per subject) and k stands for subject. 

Equation 1:

(∑_1^Nj▒〖P_jk (1-P_jk)〗)/N_j =X_k

X_k is the average of choice variance. We first calculated the variance across the repetitions per each trial type, and then calculated the average of these variances for each subject.

Equation 2:

(∑_1^Nj▒〖〖(P〗_jk-(P_k ) ®)〗^2 )/(N_j-1)=Y_k

Y_k is the variability across trial types (variance of means). We first calculated the choice probability across the repetitions per each trial type, and then calculated the variance of these choice probabilities for each subject. (P_k ) ® stands for the mean of choice proportion of option B per subject.

* We randomly chose option B. It is equivalent for both options (A and B).

If both of these measurements are indications for task engagement, we would assume a negative connection between them: the smaller the mean of choice variance per trial type, the higher the variability of the choices across trial types (which would indicate a more engaged subject). Interestingly, this is exactly what we observed when we correlated between these two measurements as shown in the figure below (Fig. 6 in S3 text): the X axis represents the mean of choice variance per trial type (X_k in equation 1), and the Y axis represents the variability of choices across trial types (Y_k in equation 2) [Experiment 1: R=-0.46, p<0.01, n=38; Replication: R=-0.63, p<0.001, n=81].”

To answer the reviewer’s concern regarding “a potential connection between task engagement in the Decoy task and the tendency to show an attraction effect”, we examined the correlation between each of these measurements for task engagement (the mean of choice variance per trial type and the variability of choices across trial types) and the choice proportion of the target. 

As shown in the figure above (Fig. 7 in S3 text), there is no significant correlation between the mean of choice variance per trial type and the choice proportion of the target in both experiments (Experiment 1: R=0.23, p=0.17, n=38; Replication: R=0.09, p=0.44, n=81; Fig. A). Additionally, there is no significant correlation between the variability of choices across trial types and the choice proportion of the target in both experiments (Experiment 1: R=-0.13, p=0.45, n=38; Replication: R=-0.03, p=0.76, n=81; Fig. B). These results indicate that subjects who had a higher variance in their choices per trial type or a small variability across trial types, and thus were probably less engaged in the Decoy task, did not choose systematically the target more or less often. 

Moreover, in order to test if there is a connection between the level of task engagement in both tasks, we examined the correlation between each of the two measurements for task engagement in the Decoy task (the mean of choice variance per trial type and the variability across trial types) and the measurement of task engagement for the Gestalt task (the slope of the logistic fit). We found no significant correlation between the mean of choice variance per trial type in the Decoy task and the Gestalt slope in both experiments (Experiment 1: R=0.16, p=0.32, n=38; Replication: R=-0.12, p=0.27, n=81; Fig. A) as well as no significant correlation between the variability across trial types in the Decoy task and the Gestalt slope (Experiment 1: R=-0.18, p=0.28, n=38; Replication: R=0.09, p=0.42, n=81; Fig. B), which demonstrates that subjects who were less engaged in the Decoy task were not necessarily less engaged in the Gestalt task as well. Hence, this strengthens our interpretation regarding the relation that we found between the two tasks and indicates that it is not caused by the lack of task engagement.

Finally, we ran a regression analysis which includes VD, Gestalt threshold, and the task engagement measurements (Gestalt slope for the Gestalt task and the mean of choice variance per trial type for the Decoy task) as predictors to the choice proportion of the target for both experiments in order to examine if there is any overall influence of task engagement on our main results [we did not include the variability of choices across trial types as a predictor since there is a significant correlation between it and the mean of choice variance per trial type].

Table 3. Summary of the mixed effects logistic regression model for variables predicting the choice proportion of the target including task engagement measurements. 

 Experiment 1 (n = 38) Replication (n = 81)

Fixed-effect Parameters B SE# Z p-val B SE# Z p-val

Constant 0.40 0.18 2.23 <.05* 0.44 0.12 3.78 <.001***

Value distance -0.25 0.14 -1.86 .06 -0.17 0.09 -1.82 .07

Gestalt Threshold -0.04 0.02 -2.06 <.05* -0.03 0.01 -2.69 <.01**

Mean of choice variance per trial type 0.26 0.54 0.48 .63 -0.22 0.35 -0.64 .52

Gestalt Slope 0.03 0.17 0.15 .88 -0.03 0.09 -0.36 .72

Random-effects Parameters var var

Constant 0.00 0.00 

Value distance 0.14 0.16 

# Robust Std. Err. (Errors clustered by Subject); * p<.05 **p<.01 *** p<.001

As presented in Table 3 (Table 2 in S3 text), none of the task engagement measurements had a significant effect on the choice proportion of the target in both experiments. Furthermore, the coefficients of our main predictors (Gestalt threshold and VD) of the model which includes the task engagement measurements (Table 3) were very similar to the coefficients of our main predictors in our main model in the paper (Table 4 in the main text) in both experiments. This result suggests that the effect of the sensitivity to the proximity low on the choice proportion of the target is not related to task engagement.

In sum, these results demonstrate that the significant negative link between the Gestalt thresholds and the choice proportion of target is not merely due to the level of engagement in both tasks.”

Additional minor comments:

1) The authors mention on multiple occasions that there are substantial individual differences and that about a quarter of their sample show the opposite of the attraction effect, the repulsion effect. The presence of individuals on the "other" side of the effect is more-or-less a necessity; If these individuals were not there, the observed effect size would be substantially greater. If the authors want to make this claim, they would need to run tests on an individual level (e.g., a binomial test). Most likely, only few of those people (if any) will show a significant deviation from a 50% chance of choosing the target option.

This is an important comment, since one of the main claims of the paper is that there are individual differences and thus, we aim to show the results at the individual level. To address this comment, we added additional analyses in the main text in the results section of Experiment 1 (page 16, line 397):

“Although the average effect across subjects is significant, it is a rather small effect. This is probably because there is a considerable heterogeneity across subjects in their probability to choose the target (the range of probabilities spreads between 0.38 and 0.68 (Fig 4A)). Therefore, additionally, we examined separately for each subject, the effect of adding a decoy on their probability to choose the target option using a binomial test. We found that only ~20% of subjects chose the target significantly different than 50% (Experiment 1: 7 out of 38 subjects (18%) chose the target significantly different than 50% (p<0.05); detailed individual results are available in S6 Appendix). While most of the subjects who showed a significant decoy effect displayed an attraction effect, 29% of them displayed the opposite effect (a repulsion effect – higher probability to choose the competitor when the decoy was asymmetrically dominated by the target [37, 38]). These results are in line with previous studies which posited that decoy effects are usually weak effects [40, 54] and that there are considerable differences between subjects [54].”

Additionally, we added these analyses in the results section of the Replication experiment (page 24, line 564):

“Additionally, there was a high variability across subjects in their probability to choose the target (the range of probabilities spreads between 0.41 and 0.83 (Fig 4B)). Similar to Experiment 1, ~20% of the subjects displayed a significant decoy effect on an individual level (17 out of 81 subjects (21%) chose the target significantly different than 50% (p<0.05); detailed individual results are available in S6 Appendix). Moreover, similarly to Experiment 1, most of the subjects who showed a significant decoy effect displayed an attraction effect, while 18% displayed a repulsion effect.”

Moreover, we added this explanation in the discussion (page 28, line 676):

“We observed, in both experiments, that only ~20% of the subjects displayed a significant decoy effect on an individual level. It is important to note that in most previous studies, only group effects were described [1-3, 14], either because the study was a between-subject’s design or because the study only focused on group effects. However, studies that did examine and report results at the individual level show that there are systematic differences across subjects in regard to the influence of context on their behavior [54, 55, 40] and posit that decoy effects are usually weak effects [40, 54] similar to our results.”

It is important to note that although the small attraction effect size and the large variability between subjects, we were able to show a significant attraction effect across subjects in both experiments, as well as we were able to explain the tendency to choose the target using the Gestalt threshold.

 As a robustness check, the authors checked for violations of WASRP. A correlation of r=.99 suggests that the main analysis and the supplemental analysis are almost identical. However, I do not see how these analyses differ from each other. Does either of the analyses include trials on which the decoy was chosen? It would be helpful if the authors could clarify the differences between the analyses.

We thank Reviewer 1 for this comment. We understand that these analyses were unclear in the original text. Therefore, we added additional explanation in the main text (the exact explanation as well as the location in the text are described further in this response) as well as detailed analyses in S3 text under ‘robustness analyses’ section in order to clarify our purpose.

We used mixed-effect logistic regressions in our main analyses, which requires a binary dependent variable per trial. This is the reason that we used the choice of either target or competitor as a measurement per trial for each subject and the choice proportion of the target overall trials (excluding trials in which the decoy option was chosen – 1% of the trials) as a measurement per subject.

The purpose of the robustness analyses was to examine if our measurement (choice proportion of the target) is consistent with other measurements that are usually applied in other studies that examined decoy effects. We wanted to discard the possibility that our main result is due to the specific measurement we chose. Therefore, we measured the attraction effect with additional three measurements: violation of WASRP (the Weak Axiom of Stochastic Revealed Preference) (Bandyopadhyay et al., 1999; Castillo, 2020), violation of regularity (Tversky, 1972) and relative choice share of the target (RST) (Berkowitsch et al., 2014; Spektor et al., 2018). The comparisons of WASRP violation and RST with our measurement (choice proportion of target) indicated that these three measurements are very similar to each other. This assured us that our measurement is valid and consistent with other measurements in the literature. 

It is important to note that both WASRP violation and RST allow a combination of both options A and B as a target, while the regularity violation do not. This is because there is no definition of a target option in the Basic condition. Therefore, we compared our measurement (choice proportion of target) only to WASRP violation and RST. Nonetheless, we present in the robustness analyses (in S3 text) the measurement of the attraction effect using the regularity violation with our data.

These are the additional explanations that we added in the main text (page 12, line 311):

“We are aware that there are other measurements for the attraction effect, however we chose specifically this one since our main analyses in which we used mixed-effect logistic regressions required a dependent variable per trial (we used the choice in each trial: target/competitor for each subject). We included analyses of the attraction effect using three other measurements that are used in the literature: violation of WASRP (the Weak Axiom of Stochastic Revealed Preference) [39, 40], violation of regularity [6] and relative choice share of the target [38]. We concluded that most of the measurements are very similar to the one we used, and thus, yield similar results (see S3 text; Robustness analyses for more details).”

 And (page 17, line 410):

“Moreover, we examined the robustness of our measurement for the attraction effect size (choice proportion of target) by comparing it with three other measurements that are used in the literature: violation of WASRP (the Weak Axiom of Stochastic Revealed Preference) [39, 40], violation of regularity [6] and relative choice share of the target [38]. We concluded that most of the measurements are very similar to the one we used, and thus, yield similar results [detailed information and analyses are provided in S3 Text].” 

Additionally, as was mentioned above we described in detail the additional analyses in S3 text under ‘robustness analyses’ section.

 In the basic condition, option A seems to be preferred to option B (60% vs. 40%, Fig. 5 in S3), at least on average. Since this is not mentioned anywhere in the text I have to guess, but I assume that A is the safe option and B is the riskier option. Moreover, there are substantial differences in the choice proportions, such that the values range from 0% to about 80%, whereas they should be closer to 50%. The authors should at least mention and briefly discuss that their calibration procedure did not achieve the desired result.

We appreciate the reviewer’s comment. First of all, in the method section (line 196), we clearly describe the attributes (amount and probability) of both A and B. Based on this description, it is clear that indeed option A is the safer option. However, as requested, we added further analyses and clarifications of the basic preference in the supplementary material (S3 text under “Higher choice proportion of the safer option” section):

“It is important to note that although we aimed to reach an indifference between options A and B using the Calibration task, the safer option (option A) was chosen more often across subjects in the Basic condition in both experiments, albeit only significant in the Replication experiment (Experiment 1: mean choice proportion of option A: 0.61, t(38)=1.89, p=0.07; Replication: mean choice proportion of option A: 0.57, t(81)=2.85, p<0.01; Fig. 5 and Fig. 9 (figure below)). There is a very wide range of preferences of option A in both experiments (Experiment 1: from 0.125 to 1, Replication: from 0 to 1; Fig 9 (figure below)). Additionally, we conducted a binomial test to examine the significance of the preference towards a specific option per subject. In both experiments, ~30% of the subjects significantly preferred one of the options (Experiment 1: 26% of the subjects significantly preferred option A over B and none of the subjects significantly preferred option B; Replication: 32% of the subjects significantly preferred one of the options, and 70% out of these subjects significantly preferred option A over B). This indicates that although we aimed for subjects to be indifferent in the subjective value between option A and B by using the Calibration task, around third of the subjects had a significant difference in the subjective value between the options. A previous study demonstrated that an increase in the subjective value difference between the relevant options (A and B) leads to a decrease in the attraction effect [10]. Therefore, this could be one of the reasons for the small attraction effect sizes in our study. Nonetheless, although the calibration task did not work perfectly, we were able to show a significant attraction effect across subjects as well as a significant link between the choice proportion of the target and the susceptibility to group by proximity.”

Additionally, we mention this in the main text in the discussion (page 28, line 683):

“Furthermore, although we aimed to reach for each subject an indifference between options A and B using the Calibration task, the safer option (option A) was chosen more often across subjects in the Basic condition in both experiments, albeit only significant in the Replication experiment. Additionally, for around third of the subjects in both experiments there was a significant difference in the subjective value between the two options even though we used a Calibration task. This could be another reason for the small attraction effect sizes in our study. Nonetheless, although the calibration task did not work perfectly, we were able to show a significant attraction effect across subjects as well as a significant link between the choice proportion of the target and the susceptibility to group by proximity.” 

Concerns raised by Reviewer 2:

Comments

This is my first review of the paper “Attraction to similar options: the Gestalt law of proximity is related to the attraction effect” by Izakson et al.. In their study, the authors investigate if and how common processes underlying perceptual and value-based decisions might cause the attraction effect. In two studies, one of them a pre-registered replication study, they basically found a correlation between susceptibility to the Gestalt law of proximity and the size of the attraction effect. While this is an interesting finding that should be published, the authors thus far did not aim at understanding why this correlation exists. I therefore recommend to do additional analyses. I furthermore feel uncomfortable with the number of excluded participants and the reasons for exclusion. Please find my details comments below.

 Please define spatial context.

We added an explanation for spatial context in the main text (page 2, line 60): “Other available or unavailable alternatives in the current environment of the choice set are considered spatial context.”

 I would not call attraction effect, compromise effect and similarity effect “decoy effects” but “context effects”.

We apologize if this term is confusing. However, this is the standard term used in the literature in many studies (Tsuzuki & Guo, Proc Annu Conf Cogn Sci Soc, 2004; Pettibone & Weddel, Organ Behav Hum Decis Process, 2000; Pettibone & Weddel, J Behav Dec Making, 2007; Marini & Paglieri, BEHAV PROCESS, 2019; Dumbalska, Li, Tsetsos & Summerfield, https://psyarxiv.com/p85mb, 2020). Our goal is to use a term that differentiates these three effects (attraction effect, compromise effect, and similarity effect) from other context effects like framing effect or temporal effects. We consider the term “decoy effects” as the appropriate term since in the field of judgment and decision making, this term is usually used to describe solely these three effects (context effects that happen as a result of an addition of a decoy option). For example:

“These effects all occur with the addition of a third alternative, called the decoy, to a two-alternative choice set and are all called decoy effects.” (page 1351 from Tsuzuki & Guo, Proc Annu Conf Cogn Sci Soc, 2004)

Therefore, as to stay loyal to the common terminology in the field we kindly request to keep and use this term.

 Line 114: Transformation into subjective scale: please use a “weaker” formulation. At least for value this notion is based on models and not on evidence. Other models take objective values as inputs without transforming them into a subjective scale.

We edited this sentence according to this comment (line 115): “computational models, in both sensory perception and value processes, use transformation of information from objective magnitudes to a subjective scale in order to explain subjects’ performance.”

 “Contemporary decision making theory” is a very unusual term in decision science.

We apologize for the confused term. We meant to use this term to oppose recent decision models from standard rational choice theories. In order to clarify our meaning, we changed this term to “suggested computational models” in line 86, and to “recent decision models” in line 125.

 More than 20% of the participants were excluded, which obviously influences the results. However, I do not see obvious “mistakes” resulting in exclusion but relevant behavior. Some people might not be able to perform well in the Gestalt task. Yes, if they chose “different” in identical trials they might have had a prior. But many other participants might have had, too. Perhaps they chose different when the distance was 20.5 pixels just because they had a prior and not because they perceive a difference. I do not see the 50% threshold as a valid exclusion criterion. Similarly, the slope of the logistic regression. Obviously, these people exist. Why should their behavior not map to behavior in the decoy task? How was the “more than 96% criterion chosen”? Also these participants not necessarily make mistakes but show their preferences. I would wish to see analyses including all participants.

We thank the Reviewer for this important comment. The exclusion criteria were a major issue in the study and was one of the main reasons that we conducted a replication experiment. We first performed a thorough exploration of our data from Experiment 1 and based all of our exclusion criteria according to it. After we pre-registered the exclusion criteria, we carefully carried exactly the same exclusion criteria in the Replication experiment. Thus, the exclusion criteria were data-driven according to Experiment 1 and were performed on the data gained from the Replication experiment without looking at the data first. Therefore, we had to calculate numerical thresholds for each exclusion criterion so we would be able to exclude subjects in the Replication experiments without looking at the data. 

Moreover, following this comment, we added the following text to the main text (page 13, line 332): “Subjects were also excluded due to lack of engagement in the Decoy task. That is, if they chose the same option more than 96% of the trials at least in two out of the four blocks of the task, which indicates that they show no variation in their choices across the different trial types (which is analogous to a low slope in the Gestalt task). We chose the 96% threshold based on a thorough exploration of our data from Experiment 1, and based all of our exclusion criteria according to it. These exclusion criteria were listed in the pre-registration of the replication experiment.”

Regarding the perceptual task, since there were 192 trials, out of which 174 were between two different arrays and 18 were between two identical arrays, responding “different” all the time would allow the subject to have 90% accuracy without being truly engaged in the task (it is important to note that the subjects did not know about this proportion). In order to exclude this kind of subjects, we had to exclude subjects according to their accuracy in the identical trials (which we referred to as “catch” trials). We decided specifically on the 50% threshold, since this is the chance level. Therefore, a subject who was wrong in at least 50% of the identical trials was considered being at chance level and biased towards answering “different” or just not paying attention to the task. It is important to note, that there were only a few subjects who showed such a behavior in both experiments (Experiment 1: 3 out of 52 subjects; Replication: 4 out of 102 subjects).

Regarding the exclusion criterion that was based on the slope of the logistic regression, it was meant to exclude subjects that were not engaged in the task. We considered subjects who had a uniform distribution of errors across the different difficulty levels in the gestalt task (different interval increases), as if they are not engaged in the task. This is especially true in psychophysical tasks where there is a true correct answer. The lower the slope of the logistic function, the more uniform are the error rates across the different difficulties the subject has. Moreover, if the slope is negative then that means that the subject had more errors in the easier trials than in the harder trials, which makes it very hard to believe that he/she was truly engaged in the task. 

Such lack of consistency in the task is a sign of almost complete disinterest in the task or a lack of understanding of the task’s instructions. Therefore, we inferred that the results of such subjects do not represent their true sensitivity in the perceptual task. Moreover, in Experiment 1 (which was the experiment we based these exclusion criteria on), most of the subjects who were excluded due to the 50% or less of accuracy in the identical trials, were also excluded due to the slope criterion (3 out of 4 subjects showed both behaviors). Additionally, their thresholds were extremely different than the thresholds of other subjects as can be seen in the following figure (outliers with thresholds of -61.42, -13.78, -20.21 and 1.21 are presented as red dots while the other subjects are presented as blue dots). Thresholds which are lower than 0 actually make no sense in a psychophysical task. This indicates that our exclusion criteria relate to each other, meaning that most subjects who showed one problematic feature, usually also showed others.

Regarding the Decoy task, the 96% threshold was chosen based on Experiment 1. As was also mentioned in the first response for Reviewer 1, it was more complicated to define an error in the Decoy task since there is no correct answer in each trial (except for the first order stochastically dominated trials in which all subjects, except one, chose the 100% winning probability options all the time). Therefore, during our exploratory analysis of Experiment 1 only, we first investigated the pattern of choices for each subject using decision trees. Using this method, we noticed that subjects who chose the same option during most of the task were driven by variables such as the number of the trial, number of block or RT. Those subjects usually had a different strategy at the beginning of the task, but then, from a particular block (each subject from a different block) it seems like they got tired (or bored) and started to choose only according to a specific strategy (safer option/larger amount, etc.) and much faster (seems like they did not even consider the other options in the choice set). This examination led us, similarly to what Reviewer 1 have suggested and analogous to the slope of the Gestalt task, to define subjects whose behavior (choice) did not change across the different trials as not engaged in the task. Using the decision trees, we were able to identify these subjects (10 problematic subjects). 

However, as we mentioned in the beginning of this response, we had to calculate a numerical threshold in order to exclude subjects in the Replication experiment without looking at the data. Therefore, we tried different thresholds ranging from 95%-97% of choosing the same option throughout the task. We found that the interval of 95.4%-96.5% yielded the same problematic subjects which we identified using the decision trees. We then, chose the 96% as our threshold since this is the mean of this interval. After collecting the data of the Replication experiment, we applied the 96% numerical threshold in order to exclude subjects in the decoy task without looking at the data. 

It is important to note that since our goal in the Replication experiment was to exclude subjects without looking at the data, we had no control over the number of excluded subjects. Therefore, for the Replication experiment, we recruited more subjects than was needed based on a power analysis and stated the number of subject that we are going to recruit a-priori in our pre-registration document. 

Although there could be other criteria for exclusion, we believe that these data-driven exclusion criteria are the right approach for this kind of experiment. Nonetheless, we understand Reviewer’s 2 concern regarding the number of subjects, which were excluded, and this is one of the main reasons we ran a replication experiment. 

Regarding the reviewer’s request to run the analyses including the excluded subjects, we approach this in the following way:

First, we must posit that the exclusion criterion in the Decoy task is more problematic than the exclusion criteria in the Gestalt task. In the Gestalt task there is an objective correct answer for each trial, therefore it is easier to identify subjects who were not engaged in the task (as was explained above). Therefore, we distinguish between the exclusion criteria in both of these tasks. We first examine our main results including only the problematic subjects from the Decoy task (where the exclusion criterion was much trickier to identify) [Table 4] and then we also excluded the problematic subjects from the Gestalt task (where the exclusion criteria are more straightforward) [Table 5]. This analysis is equivalent to the analysis conducted in Table 4 in the main text (page 19) where we used mixed-effects logistic regression with value distance and Gestalt threshold as predictors to predict choice proportion of target.

Table 4. Summary of the mixed effects logistic regression model for variables predicting the choice proportion of the target including the problematic subjects from the Decoy task (Experiment 1: 10 problematic subjects in addition to the 38; Replication: 13 problematic subjects in addition to the 81). 

 Experiment 1 (n = 48) Replication (n = 94)

Fixed-effect Parameters B SE# Z p-val B SE# Z p-val

Constant 0.38 0.13 2.86 <.01** 0.33 0.08 4.33 <.001***

Value distance -0.22 0.13 -1.78 .07 -0.16 0.09 -1.85 .06

Gestalt Threshold -0.03 0.01 -1.79 .07 -0.03 0.01 -2.61 <.01**

Random-effects Parameters var var

Constant 0.03 0.00 

Value distance 0.16 0.13 

# Robust Std. Err. (Errors clustered by Subject); * p<.05 **p<.01 *** p<.001

As shown in Table 4, when we conducted the main analysis while including the subjects that were excluded due to poor task engagement in the Decoy task, we still observed a marginal negative significant effect of the Gestalt threshold in Experiment 1 (β=-0.03, p=0.07) and a significant effect in the Replication experiment (β=-0.03, p<0.01). We also found a marginal significant effect of the value distance in both experiments (Experiment 1: β=-0.22, p=0.07; Replication: β=-0.16, p=0.06). These results indicate that our main effect (the negative effect of the Gestalt threshold on the choice proportion of target) exists even when we include the subjects, which we defined to be problematic in the Decoy task, which we strongly think were not engaged in the task.

Table 5. Summary of the mixed effects logistic regression model for variables predicting the choice proportion of the target including all problematic subjects (Experiment 1: 14 problematic subjects in addition to the 38; Replication: 21 problematic subjects in addition to the 81). 

 Experiment 1 (n = 52) Replication (n = 102)

Fixed-effect Parameters B SE# Z p-val B SE# Z p-val

Constant 0.13 0.05 2.64 <.01** 0.13 0.03 4.21 <.001***

Value distance -0.19 0.12 -1.67 .09 -0.14 0.08 -1.64 0.1

Gestalt Threshold 0.00 0.00 0.35 .73 -0.00 0.00 -0.27 .79

Random-effects Parameters var var

Constant 0.03 0.00 

Value distance 0.13 0.13 

# Robust Std. Err. (Errors clustered by Subject); * p<.05 **p<.01 *** p<.001

However, as shown in Table 5, when we included also the subjects who were poorly engaged in the Gestalt task, we have no significant effect of the Gestalt threshold or the value distance. This is not really surprising since the exclusion criteria for the Gestalt task are straightforward and by looking at the data of the problematic subjects it is obvious that they were not really engaged in the task and thus have thresholds that are extremely different than the other subjects (such as: -61.42, -13.78, -20.21) and actually makes no sense in psychophysical tasks. Thus, it is impossible to infer any connection to their Decoy results using their Gestalt thresholds. 

In sum, in any behavioral paradigm, we can find subjects who are obviously not adhering to the task but just pressing buttons to finish the task and go home. It makes no sense to use them to examine a scientific question since they do not demonstrate any valid behavior that we can understand or model. This problem is not specific to our paradigm or design. One approach to minimize the number of un-engaged subjects is to use incentive compatible designs as we used in our experiments. Another approach is to use exclusion criteria. As we stated above, we agree with Reviewer 2 that exclusion criteria can be problematic since they create a lot of degrees of freedom to the researchers. Therefore, after conducting a thorough investigation to decide on the most relevant exclusion criteria based on Experiment 1, we carefully followed the rules of pre-registration and conducted a replication experiment in order to validate our results. 

If Reviewer 2 strongly thinks that these analyses are necessary, then we can put them in the supplementary material. However, we are strongly against it, first, because of all the reasons mentioned above, and second, since it defies the whole idea of a replication study in which the exclusion criteria are decided beforehand.

 Criterion for attraction effect is >50% choice probability for target. Given some noise, I would always expect some individuals above or below 50%, which is not necessarily a sign for the attraction or repulsion effect.

This is an important comment, and similarly as we responded to Reviewer’s 1 first minor comment, we added analyses and clarifications in the main text.

To address this comment, we added additional analyses in the main text in the results section of Experiment 1 (page 16, line 397):

“Although the average effect across subjects is significant, it is a rather small effect. This is probably because there is a considerable heterogeneity across subjects in their probability to choose the target (the range of probabilities spreads between 0.38 and 0.68 (Fig 4A)). Therefore, additionally, we examined separately for each subject, the effect of adding a decoy on their probability to choose the target option using a binomial test. We found that only ~20% of subjects chose the target significantly different than 50% (Experiment 1: 7 out of 38 subjects (18%) chose the target significantly different than 50% (p<0.05); detailed individual results are available in S6 Appendix). While most of the subjects who showed a significant decoy effect displayed an attraction effect, 29% of them displayed the opposite effect (a repulsion effect – higher probability to choose the competitor when the decoy was asymmetrically dominated by the target [37, 38]). These results are in line with previous studies which posited that decoy effects are usually weak effects [40, 54] and that there are considerable differences between subjects [54].”

Additionally, we added these analyses in the results section of the Replication experiment (page 24, line 564):

“Additionally, there was a high variability across subjects in their probability to choose the target (the range of probabilities spreads between 0.41 and 0.83 (Fig 4B)). Similar to Experiment 1, ~20% of the subjects displayed a significant decoy effect on an individual level (17 out of 81 subjects (21%) chose the target significantly different than 50% (p<0.05); detailed individual results are available in S6 Appendix). Moreover, similarly to Experiment 1, most of the subjects who showed a significant decoy effect displayed an attraction effect, while 18% displayed a repulsion effect.”

Moreover, we added this explanation in the discussion (page 28, line 676):

“We observed, in both experiments, that only ~20% of the subjects displayed a significant decoy effect on an individual level. It is important to note that in most previous studies, only group effects were described [1-3, 14], either because the study was a between-subject’s design or because the study only focused on group effects. However, studies that did examine and report results at the individual level show that there are systematic differences across subjects in regard to the influence of context on their behavior [54, 55, 40] and posit that decoy effects are usually weak effects [40, 54] similar to our results.”

In sum, we agree that the attraction effect is very weak, however we claim that the tendency of the subject to be affected by the decoy is not a random noise, but a systematic process and this is why we were able to show the connection to the susceptibility to group by proximity.

 Logistic regressions: Instead of focusing on significance levels of slopes, I would want to see a model comparison of models including value difference or not (e.g. based on BIC).

To address this comment, we performed likelihood ratio tests (LRT) to compare between models and added these analyses to the supplementary material: S3 text under ‘model comparisons’ section:

“First, we compared between a mixed-effects logistic regression which includes only an intercept (M0) and a mixed-effects logistic regression which includes value distance (VD) as a predictor as well (M1). As shown in Table 6 (Table 3 in S3 text), the addition of the VD as a predictor to the regression model produced a marginal significant effect in both experiments (Experiment 1: Chisq=3.29, p=0.07; Replication: Chisq=3.22, p=0.07). Moreover, the AIC score of M1 was (slightly) better than the AIC score of M0. This indicates that the addition of VD to the model increased (slightly) the goodness-of-fit.

M0: 〖P(Choice Target)〗_ij=1/(1+〖exp〗^((-() 〖β0j+ε〗_ij))

M1: 〖P(Choice Target)〗_ij=1/(1+〖exp〗^((-() 〖β0j+β1jValue distance" " +ε〗_ij))

Table 6. Summary of model comparisons to examine the significance of adding Value distance. 

 Experiment 1 (n = 38) Replication (n = 81)

models AIC logLik deviance Chisq Chi Df Pr(>Chisq) AIC logLik deviance Chisq Chi Df Pr(>Chisq)

M0 13260.74 6626.371 13252.74 28287.47 14139.74 28279.47 

M1 13259.45 6624.723 13249.45 3.295668 1 0.06946285 28286.25 14138.13 28276.25 3.21966 1 0.07275873

Moreover, we compared between a mixed-effects logistic regression which includes only VD as a predictor of the probability to choose the target (M1) and a mixed-effects logistic regression which includes both VD and Gestalt threshold as predictors (M2). As shown in Table 7 (Table 4 in S3 text), the addition of the Gestalt threshold to the regression model produced a significant effect in both experiments (Experiment 1: Chisq=5.00, p<0.05; Replication: Chisq=7.21, p<0.01), and the AIC score was lower in M2 compared to M1.

M1: 〖P(Choice Target)〗_ij=1/(1+〖exp〗^((-() 〖β0j+β1jValue distance" " +ε〗_ij))

M2: 〖P(Choice Target)〗_ij=1/(1+〖exp〗^((-() 〖β0j+β1jValue distance" + " β2 Gestalt threshold+ε〗_ij))

Table 7. Summary of model comparisons to examine the significance of adding Gestalt threshold. 

 Experiment 1 (n = 38) Replication (n = 81)

models AIC logLik deviance Chisq Chi Df Pr(>Chisq) AIC logLik deviance Chisq Chi Df Pr(>Chisq)

M1 13259.45 -6624.723 13249.45 28286.25 -14138.13 28276.25 

M2 13256.44 -6622.222 13244.44 5.002777 1 0.02530668 28281.04 -14134.52 28269.04 7.212329 1 0.00724045

Furthermore, we compared between a mixed-effects logistic regression which includes only Gestalt threshold as a predictor of the probability to choose the target (M3) and a mixed-effects logistic regression which includes both VD and Gestalt threshold as predictors (M2). As shown in Table 8 (Table 5 in S3 text), the addition of the VD to the regression model that already includes the Gestalt threshold produced a marginal significant effect in both experiments (Experiment 1: Chisq=3.43, p<0.05; Replication: Chisq=3.32, p<0.01).

M3: 〖P(Choice Target)〗_ij=1/(1+〖exp〗^((-() 〖β0j+β1jGestalt threshold" " +ε〗_ij))

M2: 〖P(Choice Target)〗_ij=1/(1+〖exp〗^((-() 〖β0j+β1jValue distance" + " β2 Gestalt threshold+ε〗_ij))

Table 8. Summary of model comparisons to examine the significance of adding Value distance as an addition to the Gestalt threshold. 

 Experiment 1 (n = 38) Replication (n = 81)

models AIC logLik deviance Chisq Chi Df Pr(>Chisq) AIC logLik deviance Chisq Chi Df Pr(>Chisq)

M3 13257.87 -6623.935 13247.87 28282.36 -14136.18 28272.36 

M2 13256.44 -6622.222 13244.44 3.427645 1 0.06411347 28281.04 -14134.52 28269.04 3.31989 1 0.06844643

These results demonstrate that using model comparisons (LRT) yielded similar results as using the significance of slopes (Wald test). The value distance has a marginal significant effect when added to either a null model which includes only an intercept as a predictor or a model which includes only the Gestalt threshold as a predictor. Additionally, the Gestalt threshold has a significant effect when added to a model that includes only the value distance as a predictor. These results strengthen our main conclusion that there is a connection between the sensitivity to the proximity law (Gestalt threshold) and the attraction effect (choice proportion of target).”

 If I understand the analyses right, the main finding is a correlation between susceptibility to the Gestalt law of proximity and the size of the attraction effect. The question that now arises is why. I would encourage the authors to aim at answering this question. At the moment two behavioral outcome variables are correlated. An interesting approach would be to identify models that predict these behavioral outcomes variables (e.g., MDFT) and see if parameters of these models are correlated. Ideally a single model can be defined predicting both behavior in the Gestalt and the decoy task, including a single parameter driving observed correlations between behavioral outcome variables.

This is a very important comment. As the reviewer described, what we found across two independent and identical experiments with a pre-registered replication, is that the lower the Gestalt sensitivity threshold of a given subject as measured in a perceptual task, the more she tends to choose the target option in a decoy task. We agree with Reviewer 2 that we lack a mechanistic explanation for this phenomenon. We tried to answer the question: how can grouping by proximity be one of the mechanisms mediating the attraction effect in the discussion by suggesting a theoretical idea. Using the evidence that there is a close interplay between selective attention and the Gestalt grouping principles, we suggest that grouping by proximity of the more similar options is what leads people to drive more attention to these similar options. This allowed us to draw a specific connection between perceptual processing (grouping by proximity) and value-based processing (comparison between lottery options). 

We agree that computational modeling in general can add to the understanding of the mechanism underlying this connection. However, since, we already have a logistic regression model that combines both of these parameters (the Gestalt threshold and the choice proportion of the target), we feel that using other more complicated models without any justification (for instance: neural data) will not significantly contribute to the understanding of the connection between these behavioral outcomes. A computational model is also only one theoretical explanation of the data, out of many possible explanations. In future work, after collecting neural data which will be relevant to these tasks, it could be beneficial to develop a more complicated computational model, which will allow the understanding of the neural mechanism underlying this connection. 

However, to address this important comment, we acknowledged it in the discussion: 

“However, note, that this is a theoretical notion that should be examined in future studies using either imaging techniques or eye movements.” (page 27, line 665)

“Future work could examine computational models that may suggest further explanation for the mechanism underlying this interesting connection between the proximity law and the attraction effect.” (page 31, line 756)

Appendix A – individual subjects’ plots - choice variance as a function of interval increase between the dots in the Gestalt task

Experiment 1

Replication

---

## [Decision Letter · Decision Letter 1]

6 Oct 2020

Attraction to similar options: the Gestalt law of proximity is related to the attraction effect

PONE-D-20-14306R1

Dear Dr. Izakson,

We’re pleased to inform you that your manuscript has been judged scientifically suitable for publication and will be formally accepted for publication once it meets all outstanding technical requirements.

Kind regards,

Tyler Davis, Ph.D.

Academic Editor

PLOS ONE

Additional Editor Comments (optional):

Reviewers' comments:

Reviewer's Responses to Questions

**Comments to the Author**

1. If the authors have adequately addressed your comments raised in a previous round of review and you feel that this manuscript is now acceptable for publication, you may indicate that here to bypass the “Comments to the Author” section, enter your conflict of interest statement in the “Confidential to Editor” section, and submit your "Accept" recommendation.

Reviewer #1: All comments have been addressed

2. Is the manuscript technically sound, and do the data support the conclusions?

Reviewer #1: Yes

3. Has the statistical analysis been performed appropriately and rigorously? 

Reviewer #1: Yes

4. Have the authors made all data underlying the findings in their manuscript fully available?

Reviewer #1: Yes

5. Is the manuscript presented in an intelligible fashion and written in standard English?

Reviewer #1: Yes

6. Review Comments to the Author

Reviewer #1: (No Response)

7. PLOS authors have the option to publish the peer review history of their article (what does this mean?). If published, this will include your full peer review and any attached files.

Reviewer #1: **Yes: **Mikhail S. Spektor

---

## [Editor Report · Acceptance letter]

8 Oct 2020

PONE-D-20-14306R1 

Attraction to similar options: the Gestalt law of proximity is related to the attraction effect 

Dear Dr. Izakson:

I'm pleased to inform you that your manuscript has been deemed suitable for publication in PLOS ONE. Congratulations! Your manuscript is now with our production department. 

Kind regards, 

on behalf of

Dr. Tyler Davis 

Academic Editor

PLOS ONE